# FEATURE EMERGENCE VIA MARGIN MAXIMIZATION: CASE STUDIES IN ALGEBRAIC TASKS

**Depen Morwani, Benjamin L. Edelman,**[*] **Costin-Andrei Oncescu,**[*]
**Rosie Zhao,**[*] **Sham Kakade**
Harvard University
`{dmorwani,bedelman,concescu,rosiezhao}@g.harvard.edu,`
`sham@seas.harvard.edu`

## ABSTRACT

Understanding the internal representations learned by neural networks is a corner-stone challenge in the science of machine learning. While there have been significant recent strides in some cases towards understanding *how* neural networks implement specific target functions, this paper explores a complementary question – *why* do networks arrive at particular computational strategies? Our inquiry focuses on the algebraic learning tasks of modular addition, sparse parities, and finite group operations. Our primary theoretical findings analytically characterize the features learned by stylized neural networks for these algebraic tasks. Notably, our main technique demonstrates how the principle of margin maximization alone can be used to fully specify the features learned by the network. Specifically, we prove that the trained networks utilize Fourier features to perform modular addition and employ features corresponding to irreducible group-theoretic representations to perform compositions in general groups, aligning closely with the empirical observations of Nanda et al. (2023) and Chughtai et al. (2023). More generally, we hope our techniques can help to foster a deeper understanding of why neural networks adopt specific computational strategies.

## 1 INTRODUCTION

Opening the black box of neural networks has the potential to enable safer and more reliable deployments, justifications for model outputs, and clarity on how model behavior will be affected by changes in the input distribution. The research area of mechanistic interpretability (Olah et al., 2020; Elhage et al., 2021; Olsson et al., 2022; Elhage et al., 2022) aims to dissect individual trained neural networks in order to shed light on internal representations, identifying and interpreting sub-circuits that contribute to the networks' functional behavior. Mechanistic interpretability analyses typically leave open the question of *why* the observed representations arise as a result of training.

Meanwhile, the theoretical literature on inductive biases in neural networks (Soudry et al., 2018; Shalev-Shwartz & Ben-David, 2014; Vardi, 2023) aims to derive general principles governing which solutions will be preferred by trained neural networks—in particular, in the presence of underspecification, where there are many distinct ways a network with a given architecture could perform well on the training data. Most work on inductive bias in deep learning is motivated by the question of understanding why networks generalize from their training data to unobserved test data. It can be non-obvious how to apply the results from this literature instead to understand what solution will be found when a particular architecture is trained on a particular type of dataset.

In this work, we show that the empirical findings of Nanda et al. (2023) and Chughtai et al. (2023), about the representations found by networks trained to perform finite group operations, can be an-alytically explained by the inductive bias of the regularized optimization trajectory towards *margin maximization*. Perhaps surprisingly, the margin maximization property alone — typically used for the study of generalization — is sufficient to *comprehensively and precisely* characterize the richly structured features that are actually learned by neural networks in these settings. Let's begin by

---

[*]These authors contributed equally to this work.

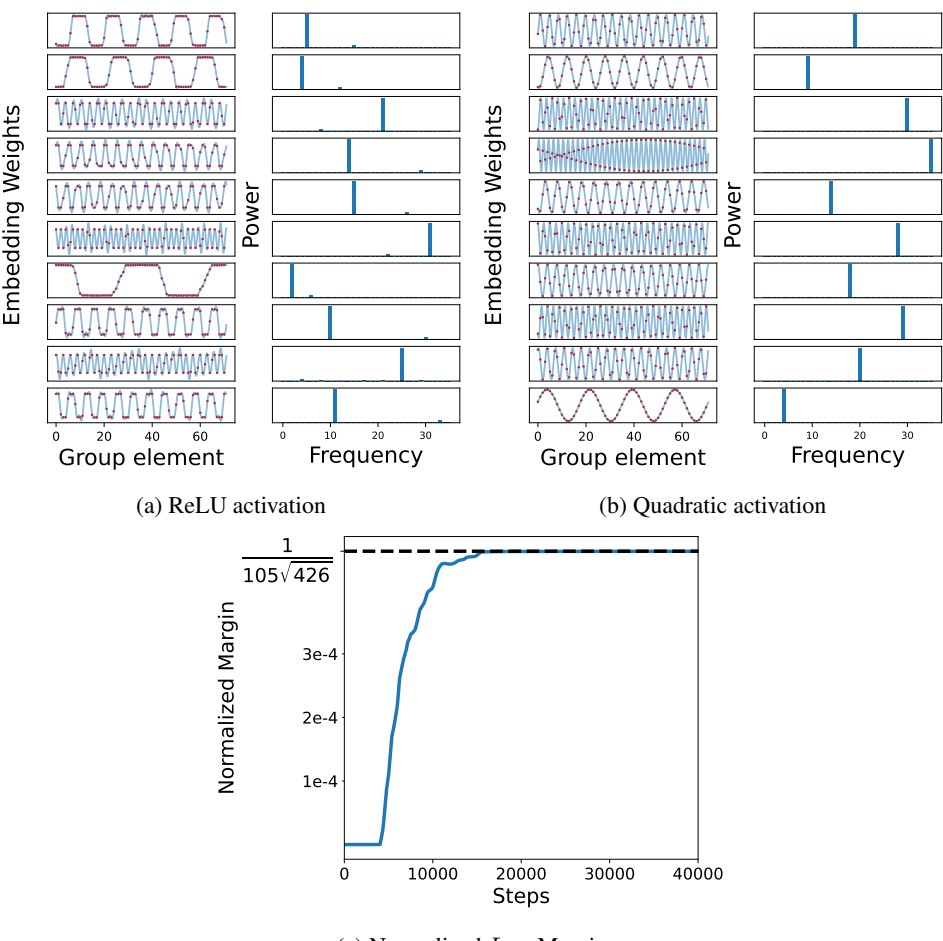

(a) ReLU activation

(b) Quadratic activation

(c) Normalized $L_{2,3}$ Margin

Figure 1: (a) Final trained embeddings and their Fourier power spectrum for a 1-hidden layer ReLU network trained on a mod-71 addition dataset with $L_2$ regularization. Each row corresponds to an arbitrary neuron from the trained network. The red dots represent the actual value of the weights, while the light blue interpolation is obtained by finding the function over the reals with the same Fourier spectrum as the weight vector. (b) Similar plot for 1-hidden layer quadratic activation, trained with $L_{2,3}$ regularization (Section 2) (c) For the quadratic activation, the network asymptotically reaches the maximum $L_{2,3}$ margin predicted by our analysis.

reviewing the case of learning modular addition with neural networks, first studied in Power et al. (2022) in their study of "grokking".

**Nanda et al.'s striking observations.** Nanda et al. (2023) investigated the problem of how neural networks learn modular addition (using a 1-layer ReLU transformer); they consider the problem of computing $a + b \mod p$, where $p$ is a prime number. The findings were unexpected and intriguing: SGD not only reliably solves this problem (as also seen in Power et al. (2022)) but also consistently learns to execute a particular algorithm, as illustrated by the learned embedding weights in Figure 1. This geometric algorithm simplifies the task to composing integer rotations around a circle. The algorithm fundamentally relies on the following identity: for any $a, b \in \mathbb{Z}_p$ and $k \in \mathbb{Z}_p \setminus \{0\}$,

$$(a + b) \bmod p = \arg\max_{c \in \mathbb{Z}_p} \left\{ \cos\left( \frac{2\pi k(a + b - c)}{p} \right) \right\}.$$

This identity leads to a few natural algorithms that are generally implemented by neural networks, as shown in Nanda et al. (2023) and Zhong et al. (2023).

These findings prompt the question: why does the network consistently prefer such Fourier-based circuits, amidst other potential circuits capable of executing the same modular addition function?

**Our Contributions.**

- We formulate general techniques for analytically characterizing the maximum margin solutions for tasks exhibiting symmetry.
- For sufficiently wide one-hidden layer MLPs with quadratic activations, we use these techniques to characterize the structure of the weights of max-margin solutions for certain algebraic tasks including modular addition, sparse parities and general group operations.
- We empirically validate that neural networks trained using gradient descent with small regularization approach the maximum margin solution (Theorem 1), and the weights of trained networks match those predicted by our theory (Figure 1).

Our theorem for modular addition shows how Fourier features are indeed the global maximum margin solution:

**Informal Theorem** (Modular addition). *Consider a single hidden layer neural network of width $m$ with $x^2$ activations trained on the modular addition task (modulo $p$). For $m \geq 4(p-1)$, any maximum margin solution satisfies the following:*

- *For every neuron, there exists a frequency such that the Fourier spectra of the input and output weight vectors are supported only on that frequency.*

- *There exists at least one neuron of each frequency in the network.*

Note that even with this activation function, there are solutions that fit all the data points, but where the weights do not exhibit any sparsity in Fourier space—see Appendix D for an example construction. Such solutions, however, have lower margin and thus are not reached by training.

In the case of $k$-sparse parity learning with an $x^k$-activation network, we show margin maximization implies that the weights assigned to all relevant bits are of the same magnitude, and the sign pattern of the weights satisfies a certain condition.

For learning on the symmetric group (or other groups with real representations), we use the machinery of representation theory (Kosmann-Schwarzbach et al., 2010) to show that learned features correspond to the irreducible representations of the group, as observed by Chughtai et al. (2023).

## 2 PRELIMINARIES

In this work, we will consider one-hidden layer neural networks with homogeneous polynomial activations, such as $x^2$, and no biases. The network output for a given input $x$ will be represented as $f(\theta, x)$, where $\theta \in \Theta$ represents the parameters of the neural network. The homogeneity constant of the network is defined as a constant $\nu$ such that for any scaling factor $\lambda > 0$, $f(\lambda\theta, x) = \lambda^\nu f(\theta, x)$.

In the case of 1-hidden layer networks, $f$ can be further decomposed as:
$f(\theta, x) = \sum_{i=1}^m \phi(\omega_i, x)$, where $\theta = \{\omega_1, \ldots, \omega_m\}$, $\phi$ represents an individual neuron within the network, and $\omega_i \in \Omega$ denotes the weights from the input to the $i^{\text{th}}$ neuron and from the neuron to the output. $\theta = \{\omega_1, \ldots, \omega_m\}$ is said to have *directional support* on $\Omega' \subseteq \Omega$ if for all $i \in \{1, \ldots, m\}$, either $\omega_i = 0$ or $\lambda_i \omega_i \in \Omega'$ for some $\lambda_i > 0$.

For Sections 4 and 6 corresponding to cyclic and general finite groups respectively, we consider neural networks with $x^2$ activation. A single neuron will be represented as $\phi(\{u, v, w\}, x^{(1)}, x^{(2)}) = (u^\top x^{(1)} + v^\top x^{(2)})^2 w$, where $u, v, w \in \mathbb{R}^d$ are the weights associated with a neuron and $x^{(1)}, x^{(2)} \in \mathbb{R}^d$ are the inputs provided to the network (note that $\phi(\{u, v, w\}, x^{(1)}, x^{(2)}) \in \mathbb{R}^d$). For these tasks, we set $d = |G|$, where $G$ refers to either the cyclic group or a general group. We will also consider the inputs $x^{(1)}$ and $x^{(2)}$ to be one-hot vectors, representing the group elements being provided as inputs. Thus, for given input elements $(a, b)$, a single neuron can be simplified as $\phi(\{u, v, w\}, a, b) = (u_a + v_b)^2 w$, where $u_a$ and $v_b$ represent the $a^{th}$ and $b^{th}$ component of $u$ and $v$ respectively. Overall, the network will be given by $f(\theta, a, b) = \sum_{i=1}^m \phi(\{u_i, v_i, w_i\}, a, b)$, with $\theta = \{u_i, v_i, w_i\}_{i=1}^m$ (note that $f(\theta, a, b) \in \mathbb{R}^d$ because there are $d$ classes).

For Section 5, we will consider the $(n, k)$-sparse parity problem, where the parity is computed on $k$ bits out of $n$. Here, we will consider a neural network with the activation function $x^k$. A single neuron within the neural network will be represented as $\phi(\{u, w\}, x) = (u^\top x)^k w$, where $u \in \mathbb{R}^n$, $w \in \mathbb{R}^2$ are the weights associated with a neuron and $x \in \mathbb{R}^n$ is the input provided to the network. The overall network will represented as $f(\theta, x) = \sum_{i=1}^m \phi(\{u_i, w_i\}, x)$, where $\theta = \{u_i, w_i\}_{i=1}^m$.

For any vector $v$ and $k \geq 1$, $\|v\|_k$ represents $\left(\sum |v_i|^k\right)^{1/k}$. For a given neural network with parameters $\theta = \{\omega_i\}_{i=1}^m$, the $L_{a,b}$ norm of $\theta$ is given by $\|\theta\|_{a,b} = \left(\sum_{i=1}^m \|\omega_i\|_a^b\right)^{1/b}$. Here $\{\omega_i\}$ represents the concatenated vector of parameters corresponding to a single neuron. For a discrete set $\mathcal{Y}$, $\Delta(\mathcal{Y})$ refers to the standard simplex over $\mathcal{Y}$.

## 3 THEORETICAL APPROACH

Suppose we have a dataset $D \subseteq \mathcal{X} \times \mathcal{Y}$, a norm $\|\cdot\|$ and a class of parameterized functions $\{f(\theta, \cdot) \mid \theta \in \mathbb{R}^U\}$, where $f : \mathbb{R}^U \times \mathcal{X} \to \mathbb{R}^{\mathcal{Y}}$ and $\Theta = \{\|\theta\| \leq 1\}$. We define the margin function $g : \mathbb{R}^U \times \mathcal{X} \times \mathcal{Y} \to \mathbb{R}$ as being, for a given datapoint $(x, y) \in D$, $g(\theta, x, y) = f(\theta, x)[y] - \max_{y' \in \mathcal{Y} \setminus \{y\}} f(\theta, x)[y']$.

Then, the margin of the dataset $D$ is given by $h : \mathbb{R}^U \to \mathbb{R}$ defined as $h(\theta) = \min_{(x,y) \in D} g(\theta, x, y)$. Similarly, we define the normalized margin for a given $\theta$ as $h(\theta/\|\theta\|)$.

We look into optimizing $\mathcal{L}_\lambda(\theta) = \frac{1}{|D|} \sum_{(x,y) \in D} l(f(\theta, x), y) + \lambda \|\theta\|^r$ where $l$ is the cross-entropy loss. Let $\theta_\lambda \in \arg\min_{\theta \in \mathbb{R}^U} \mathcal{L}_\lambda(\theta)$ and $\gamma_\lambda = h(\theta_\lambda/\|\theta_\lambda\|)$ as well as $\gamma^* = \max_{\theta \in \Theta} h(\theta)$. The following theorem by Wei et al. (2019a) argues that, when using vanishingly small regularization $\lambda$, the normalized margin of global optimizers of $\mathcal{L}_\lambda$ converges to $\gamma^*$.

**Theorem 1** (Wei et al. (2019a), Theorem 4.1). *For any norm $\|\cdot\|$, a fixed $r > 0$ and any homogeneous function $f$ with homogeneity constant $\nu > 0$, if $\gamma^* > 0$, then $\lim_{\lambda \to 0} \gamma_\lambda = \gamma^*$.*

This provides the motivation behind studying maximum margin classifiers as a good proxy for understanding the global minimizers of $\mathcal{L}_\lambda$ as $\lambda \to 0$. Henceforth, we will focus on characterizing the maximum margin solution: $\Theta^* := \arg\max_{\theta \in \Theta} h(\theta)$. Note that the maximum margin $\gamma^* = \max_{\theta \in \Theta} \min_{(x,y) \in D} g(\theta, x, y) = \max_{\theta \in \Theta} \min_{q \in \mathcal{P}(D)} \mathbb{E}_{(x,y) \sim q} [g(\theta, x, y)]$, where $\mathcal{P}(D)$ represents the set of all distributions over data points in $D$. The primary approach in this work for characterizing the maximum margin solution is to exhibit a pair $(\theta^*, q^*)$ such that

$$q^* \in \arg\min_{q \in \mathcal{P}(D)} \mathbb{E}_{(x,y) \sim q} [g(\theta^*, x, y)] \tag{1}$$

$$\theta^* \in \arg\max_{\theta \in \Theta} \mathbb{E}_{(x,y) \sim q^*} [g(\theta, x, y)] \tag{2}$$

The lemma below uses the max-min inequality (Boyd & Vandenberghe, 2004) to show that exhibiting such a pair is sufficient for establishing that $\theta^*$ is indeed a maximum margin solution. The proof for the lemma can be found in Appendix F.

**Lemma 2.** *If $(\theta^*, q^*)$ satisfies Equations 1 and 2, then $\theta^* \in \arg\max_{\theta \in \Theta} \min_{(x,y) \in D} g(\theta, x, y)$.*

We will describe our approach for finding such a pair for 1-hidden layer homogeneous neural networks for binary classification. Furthermore, we will show how exhibiting just a single pair can enable us to characterize the set of *all* maximum margin solutions. In Appendix E, we extend our techniques to multi-class classification, which is necessary for the group operation tasks.

In the context of binary classification where $|\mathcal{Y}| = 2$, the margin function $g$ for a given datapoint $(x, y) \in D$ is given by $g(\theta, x, y) = f(\theta, x)[y] - f(\theta, x)[y']$, where $y' \neq y$. For 1-hidden layer neural networks, by linearity of expectation, the expected margin is given by

$$\mathbb{E}_{(x,y) \sim q} [g(\theta, x, y)] = \sum_{i=1}^m \mathbb{E}_{(x,y) \sim q} [\phi(\omega_i, x)[y] - \phi(\omega_i, x)[y']],$$

where $y' \neq y$ and $\theta = \{\omega_i\}_{i=1}^m$. Since the expected margin of the network decomposes into the sum of expected margin of individual neurons, finding a maximum expected margin network simplifies to finding maximum expected margin neurons. Denote $\psi(\omega, x, y) = \phi(\omega, x)[y] - \phi(\omega, x)[y']$.

**Lemma 3.** *Let* $\Theta = \{\theta : \|\theta\|_{a,b} \leq 1\}$ *and* $\Theta_q^* = \arg\max_{\theta \in \Theta} \mathbb{E}_{(x,y) \sim q}[g(\theta, x, y)]$. *Similarly, let* $\Omega = \{\omega : \|\omega\|_a \leq 1\}$ *and* $\Omega_q^* = \arg\max_{\omega \in \Omega} \mathbb{E}_{(x,y) \sim q}[\psi(\omega, x, y)]$. *For binary classification:*

- ***Single neuron optimization***: *Any* $\theta \in \Theta_q^*$ *has directional support only on* $\Omega_q^*$.

- ***Combining neurons***: *If* $b = \nu$ *(the homogeneity constant of the network) and* $\omega_1^*, ..., \omega_m^* \in \Omega_q^*$, *then for any neuron scaling factors* $\sum \lambda_i^\nu = 1, \lambda_i \geq 0$, *we have that* $\theta = \{\lambda_i \omega_i^*\}_{i=1}^m$ *belongs to* $\Theta_q^*$.

The proof for the above lemma can be found in Appendix F.1.

To find a $(\theta^*, q^*)$ pair, we will start with a guess for $q^*$ (which will be the uniform distribution in our case as the datasets are symmetric). Then, using the first part of Lemma 3, we will find all neurons which can be in the support of $\theta^*$ satisfying Equation 2 for the given $q^*$. Finally, for specific norms of the form $\|\cdot\|_{a,\nu}$, we will combine the obtained neurons using the second part of Lemma 3 to obtain a $\theta^*$ such that $q^*$ satisfies Equation 1.

We think of $(\theta^*, q^*)$ as a "certificate pair". By just identifying this single solution, we can characterize the set of *all* maximum margin solutions. Denote $\text{spt}(q) = \{(x, y) \in D \mid q(x, y) > 0\}$.

**Lemma 4.** *Let* $\Theta = \{\theta : \|\theta\|_{a,b} \leq 1\}$ *and* $\Theta_q^* = \arg\max_{\theta \in \Theta} \mathbb{E}_{(x,y) \sim q}[g(\theta, x, y)]$. *Similarly, let* $\Omega = \{\omega : \|\omega\|_a \leq 1\}$ *and* $\Omega_q^* = \arg\max_{\omega \in \Omega} \mathbb{E}_{(x,y) \sim q}[\psi(\omega, x, y)]$. *For the task of binary classification, if there exists* $\{\theta^*, q^*\}$ *satisfying Equation 1 and 2, then any* $\hat{\theta} \in \arg\max_{\theta \in \Theta} \min_{(x,y) \in D} g(\theta, x, y)$ *satisfies the following:*

- $\hat{\theta}$ *has directional support only on* $\Omega_{q^*}^*$.

- *For any* $(x, y) \in \text{spt}(q^*)$, $f(\hat{\theta}, x, y) - f(\hat{\theta}, x, y') = \gamma^*$, *where* $y' \neq y$; *i.e., all points in the support of* $q^*$ *are "on the margin" for any maximum margin solution.*

The proof for the above lemma can be found in Appendix F.1.

Thus, we can say that the neurons found by Lemma 3 are indeed the exhaustive set of neurons for any maximum margin network. Moreover, any maximum margin solution will have the support of $q^*$ on the margin.

In order to extend these results to multi-class classification, we introduce the *class-weighted margin*. Consider some $\tau : D \to \Delta(\mathcal{Y})$ that assigns to every datapoint a weighting over incorrect labels. For any $(x, y) \in D$, let $\tau$ satisfy the properties that $\sum_{y' \in \mathcal{Y} \setminus \{y\}} \tau(x, y)[y'] = 1$ and $\tau(x, y)[y'] \geq 0$ for all $y' \in \mathcal{Y}$. Using this, we define the class-weighted margin $g'$ for a given datapoint $(x, y) \in D$ as

$$g'(\theta, x, y) = f(\theta, x)[y] - \sum_{y' \in \mathcal{Y} \setminus \{y\}} \tau(x, y)[y'] f(\theta, x)[y'].$$

Similarly, the class-weighted margin for a neuron is given by $\psi'(\omega, x, y) = \phi(\omega, x)[y] - \sum_{y' \in \mathcal{Y} \setminus \{y\}} \tau(x, y)[y']\phi(\omega, x)[y']$. As the class-weighted margin is linear, we can extend our techniques above to multi-class classification. For details, refer to Appendix E.

## 3.1 BLUEPRINT FOR THE CASE STUDIES

In each case study, we will want to find a network $\theta^*$ and a distribution on the input data points $q^*$, such that Equation 1 and 2 are satisfied. Informally, these are the main steps involved in the proof approach:

1. As the datasets we considered are symmetric, we consider $q^*$ to be uniformly distributed on the input data points.

2. Using **Single neuron optimization** part of Lemma 3, we find all neurons that maximize the expected margin (expected class-weighted margin for multi-class classifcation); only these neurons can be part of a network $\theta$ satisfying Equation 1.

3. Using **Combining neurons** part of Lemma 3, we construct a $\theta^*$ such that all input points are on the margin, i.e, $\theta^*$ satisfies Equation 2 (and $g' = g$ for multi-class classification).

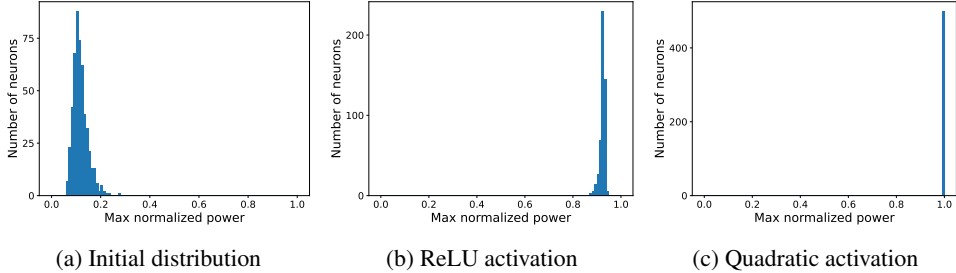

(a) Initial distribution    (b) ReLU activation    (c) Quadratic activation

Figure 2: The maximum normalized power of the embedding vector of a neuron is given by $\max_i |\hat{u}[i]|^2/(\sum |\hat{u}[j]|^2)$, where $\hat{u}[i]$ represents the $i^{th}$ component of the Fourier transform of $u$. (a) Initially, the maximum power is randomly distributed. (b) For 1-hidden layer ReLU network trained with $L_2$ regularization, the final distribution of maximum power seems to be concentrated around 0.9, meaning neurons are nearly 1-sparse in frequency space but not quite. (c) For 1-hidden layer quadratic network trained with $L_{2,3}$ regularization, the final maximum power is almost exactly 1 for all the neurons, so the embeddings are 1-sparse in frequency space, as predicted by the maximum margin analysis.

Then, using Lemma 2, we can say that the network $\theta^*$ maximizes the margin. Moreover, using Lemma 4, any margin maximizing network consists entirely of neurons found by the **Single neuron optimization** part of Lemma 3.

## 4 CYCLIC GROUPS (MODULAR ADDITION)

For a prime $p > 2$, let $\mathbb{Z}_p$ denote the cyclic group on $p$ elements. For a function $f : \mathbb{Z}_p \to \mathbb{C}$, the discrete Fourier transform of $f$ at a frequency $j \in \mathbb{Z}_p$ is defined as

$$\hat{f}(j) := \sum_{k \in \mathbb{Z}_p} f(k) \exp(-2\pi i \cdot jk/p).$$

Note that we can treat a vector $v \in \mathbb{C}^p$ as a function $v : \mathbb{Z}_p \to \mathbb{C}$, thereby endowing it with a Fourier transform. Consider the input space $\mathcal{X} := \mathbb{Z}_p \times \mathbb{Z}_p$ and output space $\mathcal{Y} := \mathbb{Z}_p$. Let the dataset $D_p := \{((a, b), a + b) : a, b \in \mathbb{Z}_p\}$.

**Theorem 5.** *Consider one-hidden layer networks $f(\theta, a, b)$ of the form given in section 2 with $m \geq 4(p - 1)$ neurons. The maximum $L_{2,3}$-margin of such a network on the dataset $D_p$ is:*

$$\gamma^* = \sqrt{\frac{2}{27}} \cdot \frac{1}{p^{1/2}(p-1)}.$$

*Any network achieving this margin satisfies the following conditions:*

1. *for each neuron $\phi(\{u, v, w\}; a, b)$ in the network, there exists a scaling constant $\lambda \in \mathbb{R}$ and a frequency $\zeta \in \{1, \ldots, \frac{p-1}{2}\}$ such that*

$$u(a) = \lambda \cos(\theta_u^* + 2\pi\zeta a/p)$$
$$v(b) = \lambda \cos(\theta_v^* + 2\pi\zeta b/p)$$
$$w(c) = \lambda \cos(\theta_w^* + 2\pi\zeta c/p)$$

   *for some phase offsets $\theta_u^*, \theta_v^*, \theta_w^* \in \mathbb{R}$ satisfying $\theta_u^* + \theta_v^* = \theta_w^*$.*

2. *Each frequency $\zeta \in \{1, \ldots, \frac{p-1}{2}\}$, is used by at least one neuron.*

*Proof outline.* Following the blueprint described in the previous section, we first prove that neurons of the form above (and only these neurons) maximize the expected class-weighted margin $\mathbb{E}_{a,b}[\psi'(u, v, w)]$ with respect to the uniform distribution $q^* = \text{unif}(\mathcal{X})$. We will use the uniform class weighting: $\tau(a, b)[c'] := 1/(p - 1)$ for all $c' \neq a + b$. As a crucial intermediate step, we prove

$$\mathbb{E}_{a,b}[\psi'(\{u, v, w\}, a, b)] = \frac{2}{(p-1)p^2} \sum_{j \neq 0} \hat{u}(j)\hat{v}(j)\hat{w}(-j),$$

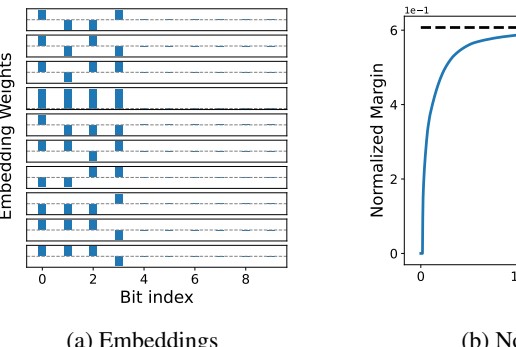

(a) Embeddings                    (b) Normalized margin

Figure 3: Final neurons with highest norm and the evolution of normalized $L_{2,5}$ margin over training of a 1-hidden layer quartic network (activation $x^4$) on $(10, 4)$ sparse parity dataset with $L_{2,5}$ regularization. The network approaches the theoretical maximum margin that we predict.

Maximizing the above expression subject to the max-margin norm constraint $\sum_{j \neq 0} \left( |\hat{u}(j)|^2 + |\hat{v}(j)|^2 + |\hat{w}(j)|^2 \right) \leq 1$ leads to sparsity in Fourier space.

Then, we describe a network $\theta^*$ (of width $4(p-1)$) composed of such neurons, that satisfies Equations 1 and 2. By Lemma 2, $\theta^*$ will be an example of a max-margin network, and part (1) of Theorem 5 will follow by Lemma 4. Finally, in order to show that all frequencies are used, we introduce the multidimensional discrete Fourier transform. We prove that each neuron only contributes a single frequency to the multi-dimensional DFT of the network; but that second part of Lemma 4 implies that all frequencies are present in the full network's multidimensional DFT. The full proof can be found in Appendix G. □

As demonstrated in Figure 1 and 2, empirical networks trained with gradient descent with $L_{2,3}$ regularization approach the theoretical maximum margin, and have single frequency neurons. Figure 5 in the Appendix verifies that all frequencies are present in the network.

## 5   SPARSE PARITY

In this section, we will establish the max margin features that emerge when training a neural network on the sparse parity task. Consider the $(n, k)$-sparse parity problem, where the parity is computed over $k$ bits out of $n$. To be precise, consider inputs $x_1, ..., x_n \in \{\pm 1\}$. For a given subset $S \subseteq [n]$ such that $|S| = k$, the parity function is given by $\Pi_{j \in S} x_j$.

**Theorem 6.** *Consider a single hidden layer neural network of width $m$ with the activation function given by $x^k$, i.e, $f(x) = \sum_{i=1}^{m} (u_i^\top x)^k w_i$, where $u_i \in \mathbb{R}^n$ and $w_i \in \mathbb{R}^2$, trained on the $(n, k)$−sparse parity task. Without loss of generality, assume that the first coordinate of $w_i$ corresponds to the output for class $y = +1$. Denote the vector $[1, -1]$ by $\boldsymbol{b}$. Provided $m \geq 2^{k-1}$, the $L_{2,k+1}$ maximum margin is:*

$$\gamma^* = k!\sqrt{2(k+1)^{-(k+1)}}.$$

*Any network achieving this margin satisfies the following conditions:*

1. *For every $i$ s.t. $\|u_i\| > 0$, $spt(u_i) = S$, $w_i$ lies in the span of $\boldsymbol{b}$ and $\forall j \in S$, $|u_i[j]| = \|w_i\|$.*

2. *For every $i$, $\left( \Pi_{j \in S} u_i[j] \right) \left( w_i^\top \boldsymbol{b} \right) \geq 0$.*

As shown in Figure 3, a network trained with gradient descent and $L_{2,k+1}$ regularization exhibits these properties, and approaches the theoretically-predicted maximum margin. The proof for Theorem 6 can be found in Appendix H.

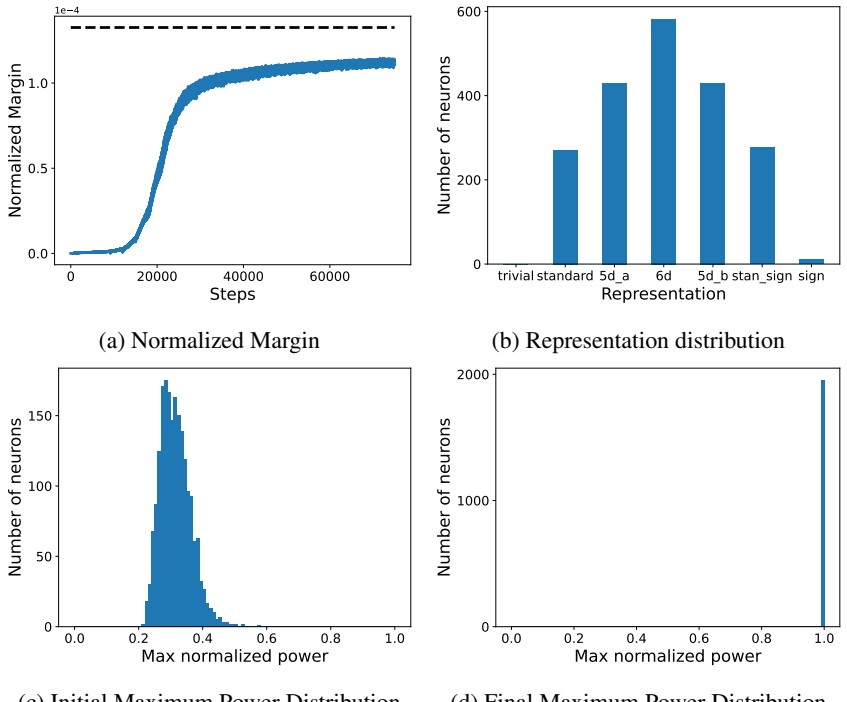

(a) Normalized Margin

(b) Representation distribution

(c) Initial Maximum Power Distribution

(d) Final Maximum Power Distribution

Figure 4: This figure demonstrates the training of a 1-hidden layer quadratic network on the symmetric group $S5$ with $L_{2,3}$ regularization. (a) Evolution of the normalized $L_{2,3}$ margin of the network with training. It approaches the theoretical maximum margin that we predict. (b) Distribution of neurons spanned by a given representation. Higher dimensional representations have more neurons as given by our construction. (c) and (d) Maximum normalized power is given by $\max_i \hat{u}[i]^2/(\sum_j \hat{u}[j]^2)$ where $\hat{u}[i]$ refers to the component of weight vector $u$ spanned by the basis vectors corresponding to $i^{th}$ representation. This is random at initialization, but towards the end of training, all neurons are concentrated in a single representation, as predicted by maximum margin.

## 6 FINITE GROUPS WITH REAL REPRESENTATIONS

We conclude our case study on algebraic tasks by studying group composition on finite groups $G$. Namely, here we set $\mathcal{X} := G \times G$ and output space $\mathcal{Y} := G$. Given inputs $a, b \in G$ we train the network to predict $c = ab$. We wish to characterize the maximum margin features similarly to the case of modular addition; here, our analysis relies on principles from group representation theory.

### 6.1 BRIEF BACKGROUND AND NOTATION

The following definitions and notation are essential for stating our main result, and further results are presented with more rigor in Appendix I.

A real representation of a group $G$ is a finite dimensional real vector space $V = \mathbb{R}^d$ and a group homomorphism (i.e. a map preserving the group structure) $R : G \to GL(V)$. We denote such a representation by $(R, V)$ or just by $R$. The dimension of a representation $R$, denoted $d_R$, is the dimension of $V$. Our analysis focuses on *unitary, irreducible, real* representations of $G$. The number of such representations is precisely equal to the number of conjugacy classes of $G$ where the conjugacy class of $a \in G$ is defined as $C(a) = \{gag^{-1} : g \in G\}$.

A quantity important to our analysis is the *character* of a representation $R$, denoted $\chi_R : G \to \mathbb{R}$ given by $\chi_R(g) = \mathrm{tr}(R(g))$. It was previously observed by Chughtai et al. (2023) that one-layer ReLU MLPs and transformers learn the task by mapping inputs $a, b$ to their respective matrices $R(a), R(b)$ for some irreducible representation $R$ and performing matrix multiplication with $R(c^{-1})$ to output logits proportional to the character $\chi_R(abc^{-1}) = \mathrm{tr}(R(a)R(b)R(c^{-1}))$, which is in par-

ticular maximized when $c = ab$. They also find evidence of network weights being *spanned by representations*, which we establish rigorously here.

For each representation $R$ we will consider the $|G|$-dimensional vectors by fixing one index in the matrices outputted by $R$, i.e. vectors $(R(g)_{(i,j)})_{g \in G}$ for some $i, j \in [d_R]$. For each $R$, this gives $d_R{}^2$ vectors. Letting $K$ be the number of conjugacy classes and $R_1, \ldots, R_K$ be the corresponding irreducible representations, since $|G| = \sum_{n=1}^K d_{R_n}^2$, taking all such vectors for each representation will form a set of $|G|$ vectors which we will denote $\rho_1, ..., \rho_{|G|}$ ($\rho_1$ is always the vector corresponding to the trivial representation). These vectors are in fact orthogonal, which follows from orthogonality relations of the representation matrix elements $R(g)_{(i,j)}$ (see Appendix I for details). Thus, we refer to this set of vectors as *basis vectors* for $\mathbb{R}^{|G|}$. One can ask whether the maximum margin solution in this case has neurons which are spanned only by basis vectors corresponding to a single representation $R$, and if all representations are present in the network— the analogous result we obtained for modular addition in Theorem 5. We show that this is indeed the case.

## 6.2 THE MAIN RESULT

Our main result characterizing the max margin features for group composition is as follows.

**Theorem 7.** *Consider a single hidden layer neural network of width $m$ with quadratic activation trained on learning group composition for $G$ with real irreducible representations. Provided $m \geq 2\sum_{n=2}^K d_{R_n}{}^3$ and $\sum_{n=2}^K d_{R_n}{}^{1.5}\chi_{R_n}(C) < 0$ for every non-trivial conjugacy class $C$, the $L_{2,3}$ maximum margin is:*

$$\gamma^* = \frac{2}{3\sqrt{3|G|}} \frac{1}{\left(\sum_{n=2}^K d_{R_n}^{2.5}\right)}.$$

*Any network achieving this margin satisfies the following conditions:*

1. *For every neuron, there exists a non-trivial representation such that the input and output weight vectors are spanned only by that representation.*

2. *There exists at least one neuron spanned by each representation (except for the trivial representation) in the network.*

The complete proof for Theorem 7 can be found in Appendix J.

The condition that $\sum_{n=2}^K d_{R_n}{}^{1.5}\chi_{R_n}(C) < 0$ for every non-trivial conjugacy class $C$ holds for the symmetric group $S_k$ up until $k = 5$. In this case, as shown in Figure 4, network weights trained with gradient descent and $L_{2,3}$ regularization exhibit similar properties. The maximum margin of the network approaches what we have predicted in theory. Analogous results for training on $S_3$ and $S_4$ in Figures 6 and 7 are in the Appendix.

Although Theorem 7 does not apply to all finite groups with real representations, it can be extended to apply more generally. The theorem posits that *every* representation is present in the network, and *every* conjugacy class is present on the margin. Instead, for general finite groups, each neuron still satisfies the characteristics of max margin solutions in that it is only spanned by one non-trivial representation, but only a *subset* of representations are present in the network; moreover, only a subset of conjugacy classes are present on the margin. More details are given in Appendix J.2.

## 7 DISCUSSION

We have shown that the simple condition of margin maximization can, in certain algebraic learning settings, imply very strong conditions on the representations learned by neural networks. The mathematical techniques we introduce are general, and may be able to be adapted to other settings than the ones we consider. Our proof holds for the case of $x^2$ activations ($x^k$ activations, in the $k$-sparse parity case) and $L_{2,\nu}$ norm, where $\nu$ is the homogeneity constant of the network. Empirical findings suggest that the results may be transferable to other architectures and norms. In general, we think explaining how neural networks adapt their representations to symmetries and other structure in data is an important subject for future theoretical and experimental inquiry.

## 8 ACKNOWLEDGMENTS

We thank Boaz Barak for helpful discussions. This work has been made possible in part by a gift from the Chan Zuckerberg Initiative Foundation to establish the Kempner Institute for the Study of Natural and Artificial Intelligence. Sham Kakade acknowledges funding from the Office of Naval Research under award N00014-22-1-2377. Ben Edelman acknowledges funding from the National Science Foundation Graduate Research Fellowship Program under award DGE-214074. Depen Morwani, Costin-Andrei Oncescu and Rosie Zhao acknowledge support from Simons Investigator Fellowship, NSF grant DMS-2134157, DARPA grant W911NF2010021, and DOE grant DE-SC0022199.

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

# Part I

# Appendix

## Table of Contents

# A   FURTHER RELATED WORK

Two closely related works to ours are Gromov (2023) and Bronstein et al. (2022). Gromov (2023) provides an analytic construction of various two-layer quadratic networks that can solve the modular addition task. The construction used in the proof of Theorem 5 is a special case of the given scheme. Bronstein et al. (2022)shows that all max margin solutions of a one-hidden-layer ReLU network (with fixed top weights) trained on read-once DNFs have neurons which align with clauses. However, the proof techniques are significantly different. For any given neural network not satisfying the desired conditions ((neurons aligning with the clauses), Bronstein et al. (2022) construct a perturbed network satisfying the conditions which exhibits a better margin. We rely on the max-min duality for certifying a maximum margin solution, as shown in Section 3.1.

**Margin maximization.** One branch of results on margin maximization in neural networks involve proving that the optimization of neural networks leads to an implicit bias towards margin maximization. Soudry et al. (2018) show that logistic regression converges in direction to the max margin classifier. Wei et al. (2019b) prove that the global optimum of weakly-regularized cross-entropy loss on homogeneous networks reaches the max margin. Similarly, Lyu & Li (2019) and Ji & Telgarsky (2020) show that in homogeneous networks, even in the absence of explicit regularization, if loss becomes low enough then the weights will tend in direction to a KKT point of the max margin optimization objective. This implies margin maximization in deep linear networks, although it is not necessarily the global max margin (Vardi et al., 2022). Chizat & Bach (2020) prove that infinite-width 2-homogeneous networks with mean field initialization will converge to the global max margin solution. In a different setting, Lyu et al. (2021) and Frei et al. (2022b) show that the margin is maximized when training leaky-ReLU one hidden layer networks with gradient flow on linearly separable data, given certain assumptions on the input (eg. presence of symmetries, near-orthogonality). For more on studying inductive biases in neural networks, refer to Vardi (2023).

Numerous other works do not focus on neural network dynamics and instead analyze properties of solutions with good margins (Bartlett, 1996). For instance, Frei et al. (2023) show that the maximum margin KKT points have "benign overfitting" properties. The works by Lyu et al. (2021), Morwani et al. (2023) and Frei et al. (2023) show that max margin implies linear decision boundary for solutions. Gunasekar et al. (2018) show that under certain assumptions, gradient descent on depth-two linear convolutional networks (with weight-sharing in first layer) converges not to the standard $L_2$ max margin, but to the global max margin with respect to the $L_1$ norm of the Fourier transform of the predictor. Our work follows a similar vein, in which we characterize max margin features in our setting and relate this to trained networks via results from Wei et al. (2019b).

**Training on algebraic tasks and mechanistic interpretability.** Studying neural networks trained on algebraic tasks has offered insights into their training dynamics and inductive biases, with the simpler setting lending a greater ease of understanding. One such example is the task of modular addition, which was studied in Power et al. (2022) in their study of grokking, leading to multiple follow-up works (Liu et al., 2022; 2023). Another example is the problem of learning parities for neural networks, which has been investigated in numerous works (Daniely & Malach, 2020; Zhenmei et al., 2022; Frei et al., 2022a; Barak et al., 2022; Edelman et al., 2023). Other mathematical tasks like learning addition have been used to investigate whether models possess algorithmic reasoning capabilities (Saxton et al., 2018; Hendrycks et al., 2021; Lewkowycz et al., 2022).

The area of mechanistic interpretability aims to understand the internal representations of individual neural networks by analyzing its weights. This form of analysis has been applied to understand the motifs and features of neurons in *circuits*— particular subsets of a neural network— in computer vision models (Olah et al., 2020; Cammarata et al., 2020) and more recently in language models (Elhage et al., 2021; Olsson et al., 2022). However, the ability to fully reverse engineer a neural network is extremely difficult for most tasks and architectures. Some work in this area has shifted towards finding small, toy models that are easier to interpret, and employing labor intensive approaches to reverse-engineering specific features and circuits in detail(Elhage et al., 2022). In Nanda et al. (2023), the authors manage to fully interpret how one-layer transformers implement modular addition and use this knowledge to define progress measures that precede the grokking phase transition which was previously observed to occur for this task (Power et al., 2022). Chughtai et al. (2023) extends this analysis to learning composition for various finite groups, and identifies analogous results and progress measures. In this work, we show that these empirical findings can be

analytically explained via max margin analysis, due to the implicit bias of gradient descent towards margin maximization.

## B EXPERIMENTAL DETAILS

In this section, we will provide the hyperparameter settings for various experiments in the paper.

### B.1 CYCLIC GROUP

We train a 1-hidden layer network with $m = 500$, using gradient descent on the task of learning modular addition for $p = 71$ for $40000$ steps. The initial learning rate of the network is $0.05$, which is doubled on the steps - $[1e3, 2e3, 3e3, 4e3, 5e3, 6e3, 7e3, 8e3, 9e3, 10e3]$. Thus, the final learning rate of the network is $51.2$. This is done to speed up the training of the network towards the end, as the gradient of the loss goes down exponentially. For quadratic network, we use a $L_{2,3}$ regularization of $1e - 4$. For ReLU network, we use a $L_2$ regularization of $1e - 4$.

### B.2 SPARSE PARITY

We train a 1-hidden layer quadratic network with $m = 40$ on $(10, 4)-$sparse parity task. It is trained by gradient descent for $30000$ steps with a learning rate of $0.1$ and $L_{2,5}$ regularization of $1e - 3$.

### B.3 GENERAL GROUPS

The hyperparameters for various groups $S_3, S_4$, and $S_5$ are provided in subsections below.

#### B.3.1 S3

We train a 1-hidden layer quadratic network with $m = 30$, using gradient descent for $50000$ steps, with a $L_{2,3}$ regularization of $1e - 7$. The initial learning rate is $0.05$, which is doubled on the steps - $[200, 400, 600, 800, 1000, 1200, 1400, 1600, 1800, 2000, 2200, 2400, 2600, 5000, 10000]$. Thus, the final learning rate is $1638.4$. This is done to speed up the training of the network towards the end, as the gradient of the loss goes down exponentially.

#### B.3.2 S4

We train a 1-hidden layer quadratic network with $m = 200$, using gradient descent for $50000$ steps, with a $L_{2,3}$ regularization of $1e - 7$. The initial learning rate is $0.05$, which is doubled on the steps - $[200, 400, 600, 800, 1000, 1200, 1400, 1600, 1800, 2000, 2200, 2400, 2600, 5000, 10000]$. Thus, the final learning rate is $1638.4$. This is done to speed up the training of the network towards the end, as the gradient of the loss goes down exponentially.

#### B.3.3 S5

We train a 1-hidden layer quadratic network with $m = 2000$, using stochastic gradient descent for $75000$ steps, with a batch size of $1000$ and $L_{2,3}$ regularization of $1e - 5$. The initial learning rate is $0.05$, which is doubled on the steps - $[3000, 6000, 9000, 12000, 15000, 18000, 21000, 24000]$. Thus, the final learning rate is $12.8$. This is done to speed up the training of the network towards the end, as the gradient of the loss goes down exponentially.

## C ADDITIONAL EXPERIMENTS

The distribution of neurons of a particular frequency for the modular addition case is shown in Figure 5. As can be seen, for both ReLU and quadratic activation, the distribution is close to uniform.

Experimental results for other symmetric groups $S_3$ and $S_4$ in Figures 6 and 7 respectively. We observe the same max margin features as stated in Theorem 7 and the $L_{2,3}$ margin approaches the theoretical max margin that we have predicted.

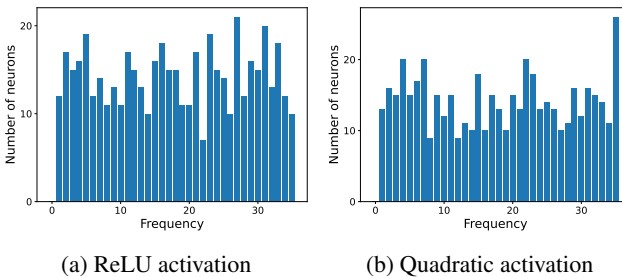

(a) ReLU activation        (b) Quadratic activation

Figure 5: Final distribution of the neurons corresponding to a particular frequency in (a) ReLU network trained with $L_2$ regularization and (b) Quadratic network trained with $L_{2,3}$ regularization. Similar to our construction, the final distribution across frequencies is close to uniform.

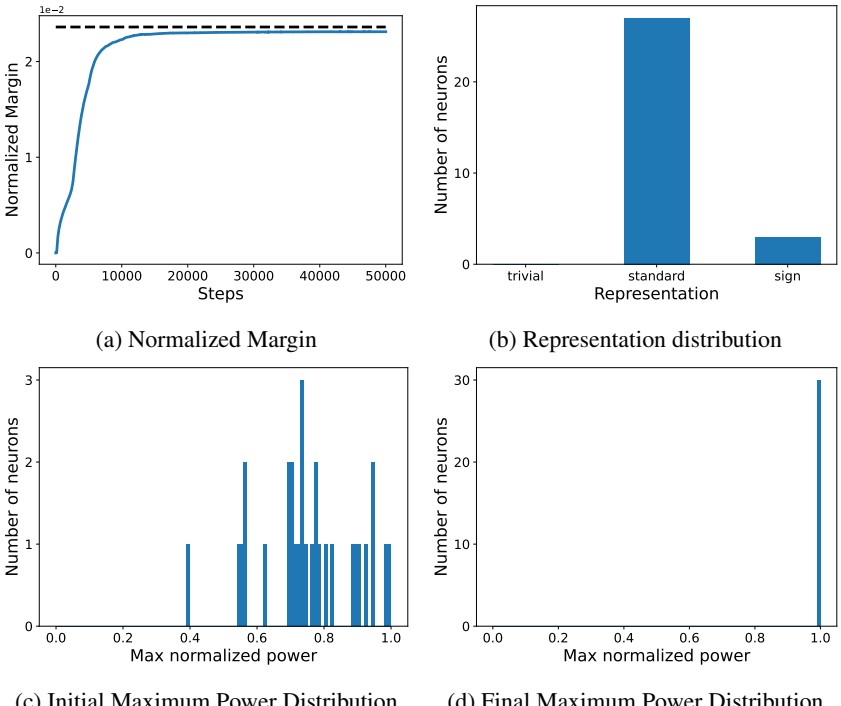

(a) Normalized Margin        (b) Representation distribution

(c) Initial Maximum Power Distribution      (d) Final Maximum Power Distribution

Figure 6: This figure demonstrates the training of a 1-hidden layer quadratic network on the symmetric group $S3$ with $L_{2,3}$ regularization. (a) Evolution of the normalized $L_{2,3}$ margin of the network with training. It approaches the theoretical maximum margin that we predict. (b) Distribution of neurons spanned by a given representation. Higher dimensional representations have more neurons as given by our construction. (c) and (d) Maximum normalized power is given by $\frac{\max \hat{u}[i]^2}{\sum_j \hat{u}[j]^2}$ where $\hat{u}[i]$ refers to the component of weight $u$ along $i^{th}$ representation. Initially, it's random, but towards the end of training, all neurons are concentrated in a single representation, as predicted by maximum margin.

## D    ALTERNATIVE CONSTRUCTION

To argue why the problem of finding correctly classifying networks is overdetermined, we present an alternative construction (which applies to general groups) that does not have an "interesting" Fourier spectrum or any behavioral similarity to the solutions reached by standard training.

For any function $r : [n]^2 \to [n]$, there exists a neural network parameterized by $\theta$ of the form considered in Sections 4 and 6 with $2p^2$ neurons such that $f(\theta, (a, b))[c] = \mathbf{1}_{c=r(a,b)}$ and that

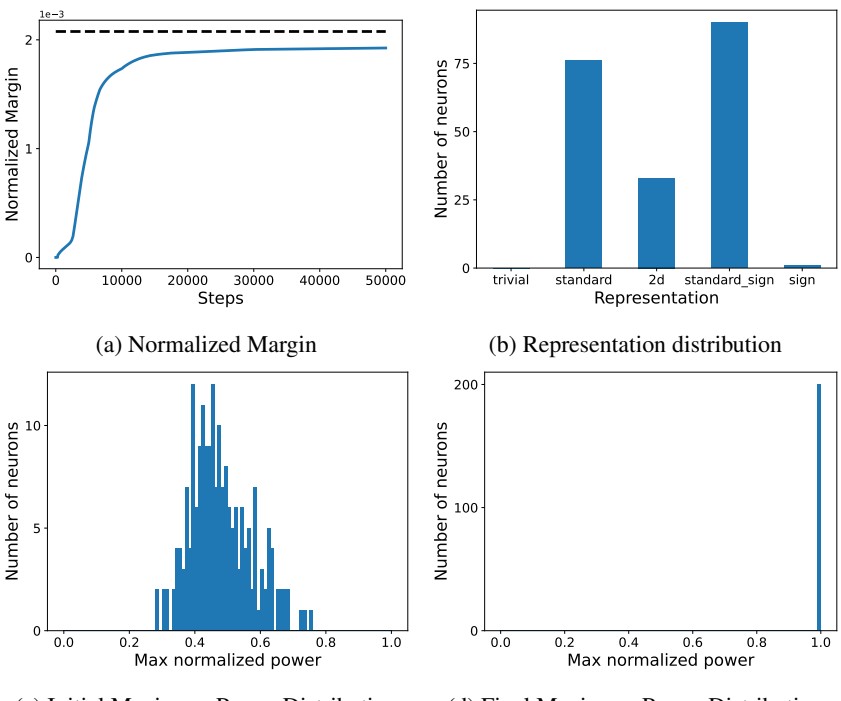

(a) Normalized Margin

(b) Representation distribution

(c) Initial Maximum Power Distribution

(d) Final Maximum Power Distribution

Figure 7: This figure demonstrates the training of a 1-hidden layer quadratic network on the symmetric group $S4$ with $L_{2,3}$ regularization. (a) Evolution of the normalized $L_{2,3}$ margin of the network with training. It approaches the theoretical maximum margin that we predict. (b) Distribution of neurons spanned by a given representation. Higher dimensional representations have more neurons as given by our construction. (c) and (d) Maximum normalized power is given by $\frac{\max \hat{u}[i]^2}{\sum_j \hat{u}[j]^2}$ where $\hat{u}[i]$ refers to the component of weight $u$ along $i^{th}$ representation. Initially, it is random, but towards the end of training, all neurons are concentrated on a single representation, as predicted by the maximum margin analysis.

is "dense" in the Fourier spectrum. For each pair $(a, b)$ we use two neurons given by $\{u, v, w\}$ and $\{u', v', w'\}$, where $u_i = u'_i = \mathbf{1}_{i=a}$, $v_i = \mathbf{1}_{i=b}$, $v'_i = -\mathbf{1}_{i=b}$, $w_i = \mathbf{1}_{i=r(a,b)}/4$ and $w'_i = -\mathbf{1}_{i=r(a,b)}/4$. When adding together the outputs for these two neurons, for an input of $(i, j)$ we get $k^{\text{th}}$ logit equal to:

$$\frac{1}{4}\left((\mathbf{1}_{i=a} + \mathbf{1}_{j=b})^2\mathbf{1}_{k=r(i,j)} - (\mathbf{1}_{i=a} - \mathbf{1}_{j=b})^2\mathbf{1}_{k=r(i,j)}\right) = \mathbf{1}_{i=a}\mathbf{1}_{j=b}\mathbf{1}_{k=r(a,b)}$$

Hence, these two neurons help "memorize" the output for $(a, b)$ while not influencing the output for any other input, so when summing together all these neurons we get an $f$ with the aforementioned property. Note that all the vectors used are (up to sign) one-hot encodings and thus have an uniform norm in the Fourier spectrum. This is to show that Fourier sparsity is not present in any correct classifier.

## E    THEORETICAL APPROACH - MULTI-CLASS CLASSIFICATION

The modular addition and general finite group tasks are multi-class classification problems. For multi-class classification, the margin function $g$ for a given datapoint $(x, y) \in D$ is given by

$$g(\theta, x, y) = f(\theta, x)[y] - \max_{y' \in \mathcal{Y} \setminus \{y\}} f(\theta, x)[y'],$$

For 1-hidden layer networks, the expected margin is given by

$$\mathbb{E}_{(x,y)\sim q}[g(\theta,x,y)] = \mathbb{E}_{(x,y)\sim q}\left[\sum_{i=1}^{m}\phi(\omega_i,x)[y] - \max_{y'\in\mathcal{Y}\setminus\{y\}}\sum_{i=1}^{m}\phi(\omega_i,x)[y']\right],$$

Here, due to the max operation, we cannot swap the summation and expectation, and thus the expected margin of the network does not decompose into the expected margins of the neurons as it did in the binary classification case.

To circumvent this issue, we will introduce the notion of *class-weighted margin*. Consider some $\tau : D \to \Delta(\mathcal{Y})$ that assigns a weighting of incorrect labels to every datapoint. For any $(x,y) \in D$, let $\tau$ satisfy the properties that $\sum_{y'\in\mathcal{Y}\setminus\{y\}}\tau(x,y)[y'] = 1$ and $\tau(x,y)[y'] \geq 0$ for all $y' \in \mathcal{Y}$. Using this, we define the class-weighted margin $g'$ for a given datapoint $(x,y) \in D$ as

$$g'(\theta,x,y) = f(\theta,x)[y] - \sum_{y'\in\mathcal{Y}\setminus\{y\}}\tau(x,y)[y']f(\theta,x)[y'].$$

Note that $g'(\theta,x,y) \geq g(\theta,x,y)$ as $g'$ replaces the max by a weighted sum. Moreover, by linearity of expectation we can say that

$$\mathbb{E}_{(x,y)\sim q}[g'(\theta,x,y)] = \sum_{i=1}^{m}\mathbb{E}_{(x,y)\sim q}\left[\phi(\omega_i,x)[y] - \sum_{y'\in\mathcal{Y}\setminus\{y\}}\tau(x,y)[y']\phi(\omega_i,x)[y']\right],$$

Denoting $\psi'(\omega,x,y) = \phi(\omega,x)[y] - \sum_{y'\in\mathcal{Y}\setminus\{y\}}\tau(x,y)[y']\phi(\omega,x)[y']$, a result analogous to Lemma 3 holds for the class-weighted margin (proof can be found in Appendix F.2):

**Lemma 8.** *Let* $\Theta = \{\theta : \|\theta\|_{a,b} \leq 1\}$ *and* $\Theta_q'^* = \arg\max_{\theta\in\Theta}\mathbb{E}_{(x,y)\sim q}[g'(\theta,x,y)]$. *Similarly, let* $\Omega = \{\omega : \|\omega\|_a \leq 1\}$ *and* $\Omega_q'^* = \arg\max_{\omega\in\Omega}\mathbb{E}_{(x,y)\sim q}[\psi'(\omega,x,y)]$. *Then:*

- ***Single neuron optimization****: Any* $\theta \in \Theta_q'^*$ *has directional support only on* $\Omega_q'^*$.

- ***Combining neurons****: If* $b = \nu$ *and* $\omega_1^*,...,\omega_m^* \in \Omega_q'^*$, *then for any neuron scaling factors* $\sum\lambda_i^\nu = 1, \lambda_i \geq 0$, *we have that* $\theta = \{\lambda_i\omega_i^*\}_{i=1}^m$ *belongs to* $\Theta_q'^*$.

The above lemma helps us characterize $\Theta_q'^*$ for a given distribution $q$. Thus, applying it to a given $q^*$, we can find

$$\theta^* \in \arg\max_{\theta\in\Theta}\mathbb{E}_{(x,y)\sim q^*}[g'(\theta,x,y)]. \tag{3}$$

To further ensure that $\theta^*$ also satisfies the corresponding equation for $g$ (i.e., Equation 2) we will consider the following condition:

**C.1** For any $(x,y) \in \text{spt}(q^*)$, it holds that $g'(\theta^*,x,y) = g(\theta^*,x,y)$. This translates to any label with non-zero weight being one of the incorrect labels where $f$ is maximized: $\{\ell \in \mathcal{Y}\setminus\{y\} : \tau(x,y)[\ell] > 0\} \subseteq \arg\max_{\ell\in\mathcal{Y}\setminus\{y\}} f(\theta^*,x)[\ell]$.

The main lemma used for finding the maximum margin solutions for multi-class classification is stated below:

**Lemma 9.** *Let* $\Theta = \{\theta : \|\theta\|_{a,b} \leq 1\}$ *and* $\Theta_q'^* = \arg\max_{\theta\in\Theta}\mathbb{E}_{(x,y)\sim q}[g'(\theta,x,y)]$. *Similarly, let* $\Omega = \{\omega : \|\omega\|_a \leq 1\}$ *and* $\Omega_q'^* = \arg\max_{\omega\in\Omega}\mathbb{E}_{(x,y)\sim q}[\psi'(\omega,x,y)]$. *If* $\exists\{\theta^*,q^*\}$ *satisfying Equations 1 and 3, and **C.1** holds, then:*

- $\theta^* \in \arg\max_{\theta\in\Theta} g(\theta,x,y)$

- *Any* $\hat{\theta} \in \arg\max_{\theta\in\Theta}\min_{(x,y)\in D} g(\theta,x,y)$ *satisfies the following:*

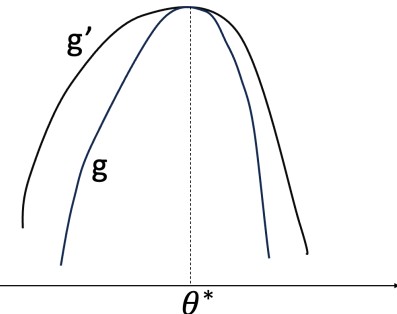

Figure 8: A schematic illustration of the relation between class-weighted margin $g'$ and maximum margin $g$.

  - $\hat{\theta}$ *has directional support only on* $\Omega'^*_{q^*}$.
  - *For any* $(x,y) \in spt(q^*)$, $f(\hat{\theta}, x, y) - \max_{y' \in \mathcal{Y} \setminus \{y\}} f(\hat{\theta}, x, y') = \gamma^*$, *i.e, all points in the support of* $q^*$ *are on the margin for any maximum margin solution.*

The first part of the above lemma follows from the fact that $g'(\theta, x, y) \geq g(\theta, x, y)$. Thus, any maximizer of $g'$ satisfying $g' = g$ is also a maximizer of $g$ (See Figure 8). The second part states that the neurons found using Lemma 8 are indeed the exhaustive set of neurons for any maximum margin network. Moreover, any maximum margin solution has the support of $q^*$ on margin. The proof for the lemma can be found in Appendix F.2.

Overall, to find a $(\theta^*, q^*)$ pair, we will start with a guess of $q^*$ (which will be uniform in our case as the datasets are symmetric) and a guess of the weighing $\tau$ (which will be uniform for the modular addition case). Then, using the first part of Lemma 8, we will find all neurons which can be in the support of $\theta^*$ satisfying Equation 3 for the given $q^*$. Finally, for specific norms of the form $\| \cdot \|_{a,\nu}$, we will combine the obtained neurons using the second part of Lemma 8 to obtain a $\theta^*$ such that it satisfies **C.1** and $q^*$ satisfies Equation 1. Thus, we will primarily focus on maximum margin with respect to $L_{2,\nu}$ norm in this work.

### E.1 BLUEPRINT FOR THE CASE STUDIES

In each case study, we want to find a certificate pair: a network $\theta^*$ and a distribution on the input data points $q^*$, such that Equation 1 and 2 are satisfied. Informally, these are the main steps involved in the proof approach:

1. As the datasets we considered are symmetric, we consider $q^*$ to be uniformly distributed on the input data points.
2. Using the **Single neuron optimization** part of Lemma 8, we find all neurons that maximize the expected class-weighted margin. Only these neurons can be part of a network $\theta^*$ satisfying Equation 3.
3. Using the **Combining neurons** part of Lemma 8, we combine the above neurons into a network $\theta^*$ such that
   (a) All input points are on the margin, i.e, $q^*$ satisfies Equation 1.
   (b) The class-weighted margin is equal to the maximum margin, i.e, $\theta^*$ satisfies **C.1**.

Then, using Lemma 9, we can say that the network $\theta^*$ maximizes the margin.

## F PROOFS FOR THE THEORETICAL APPROACH

For ease of the reader, we will first restate Equations 1 and 2.

$$q^* \in \underset{q \in \mathcal{P}(D)}{\arg\min} \underset{(x,y) \sim q}{\mathbb{E}} [g(\theta^*, x, y)]$$

$$\theta^* \in \arg\max_{\theta \in \Theta} \mathbb{E}_{(x,y)\sim q^*} [g(\theta, x, y)]$$

We will first provide the proof of Lemma 2.

**Lemma.** *If a pair $(\theta^*, q^*)$ satisfies Equations 1 and 2, then*

$$\theta^* \in \arg\max_{\theta \in \Theta} \min_{(x,y) \in D} g(\theta, x, y)$$

*Proof.* First, using max-min inequality, we have:

$$\max_{\theta \in \Theta} \min_{(x,y) \in D} g(\theta, x, y) = \max_{\theta \in \Theta} \min_{q \in \mathcal{P}(D)} \mathbb{E}_{(x,y)\sim q} [g(\theta, x, y)] \leq$$

$$\min_{q \in \mathcal{P}(D)} \max_{\theta \in \Theta} \mathbb{E}_{(x,y)\sim q} [g(\theta, x, y)]$$

On the other hand, it also holds that:

$$\min_{q \in \mathcal{P}(D)} \max_{\theta \in \Theta} \mathbb{E}_{(x,y)\sim q} [g(\theta, x, y)] \leq \max_{\theta \in \Theta} \mathbb{E}_{(x,y)\sim q^*} [g(\theta, x, y)] =$$

$$\mathbb{E}_{(x,y)\sim q^*} [g(\theta^*, x, y)] = \min_{q \in \mathcal{P}(D)} \mathbb{E}_{(x,y)\sim q} [g(\theta^*, x, y)] \leq$$

$$\max_{\theta \in \Theta} \min_{q \in \mathcal{P}(D)} \mathbb{E}_{(x,y)\sim q} [g(\theta, x, y)]$$

where the first equality follows from Equation 2 and the second follows from Equation 1. Putting these inequalities together it follows that all of the above terms are equal (and, thus we get a minimax theorem). In particular, $\theta^* \in \arg\max_{\theta \in \Theta} \min_{(x,y) \in D} g(\theta, x, y)$ as desired. $\square$

### F.1 BINARY CLASSIFICATION

Now, we will provide the proof of Lemma 3.

**Lemma.** *Let $\Theta = \{\theta : \|\theta\|_{a,b} \leq 1\}$ and $\Theta_q^* = \arg\max_{\theta \in \Theta} \mathbb{E}_{(x,y)\sim q} [g(\theta, x, y)]$. Similarly, let $\Omega = \{\omega : \|\omega\|_a \leq 1\}$ and $\Omega_q^* = \arg\max_{\omega \in \Omega} \mathbb{E}_{(x,y)\sim q} [\psi(\omega, x, y)]$. Then, for binary classification, the following holds:*

- ***Single neuron optimization:** Any $\theta \in \Theta_q^*$ has directional support only on $\Omega_q^*$.*

- ***Using multiple neurons:** If $b = \nu$ and $\omega_1^*, ..., \omega_m^* \in \Omega_q^*$, then $\theta = \{\lambda_i \omega_i^*\}_{i=1}^m$ with $\sum \lambda_i^\nu = 1, \lambda_i \geq 0$ belongs to $\Theta_q^*$.*

*Proof.* Let $\gamma = \max_{\omega \in \Omega} \mathbb{E}_{(x,y)\sim q^*} [\psi(\omega, x, y)]$ and take any $\theta = \{\omega_i\}_{i=1}^m$. Then:

$$\mathbb{E}_{(x,y)\sim q^*} [g(\theta, x, y)] = \mathbb{E}_{(x,y)\sim q^*} \left[ \sum_{i=1}^m \psi(\omega_i) \right] = \sum_{i=1}^m \|\omega_i\|_a^\nu \mathbb{E}_{(x,y)\sim q^*} \left[ \psi \left( \frac{\omega_i}{\|\omega_i\|_a} \right) \right] \leq$$

$$\gamma \sum_{i=1}^m \|\omega_i\|_a^\nu \leq \gamma \max_{\substack{w \in \mathbb{R}^m \\ \|w\|_b \leq 1}} \|w\|_\nu^\nu$$

with equality when $\frac{\omega_i}{\|\omega_i\|_a} \in \arg\max_{\omega \in \Omega} \mathbb{E}_{(x,y)\sim q^*} [\psi(\omega, x, y)]$ for all $i$ with $\omega_i \neq 0$ and the $L_a$ norms of $\omega$s respect $\{\|\omega_i\|_a\}_{i=1}^m \in \arg\max_{\|w\|_b \leq 1} \|w\|_\nu^\nu$. Since there exists equality for this upper bound, these two criteria define precisely $\arg\max_{\theta \in \Theta} \mathbb{E}_{(x,y)\sim q^*} [g(\theta, x, y)]$. Hence, we proved the first part of the statement by first criterion. For the second, note that when $b = \nu$, one can choose any vector of norms for $\omega$ with $L_b$ norm of 1 (since $\|w\|_\nu^\nu = \|w\|_b^b \leq 1$), such as $\lambda$ - this concludes the proof of the second part. $\square$

**Remark.** *Note that the analysis in above proof can be used to compute optimal norms for $b \neq \nu$ as well - however, for any such $b$ we would not get the same flexibility to build a $\theta^*$ satisfying Equation 1. This is the reason behind choosing $b = \nu$.*

Now, we will provide the proof of Lemma 4.

**Lemma.** *Let $\Theta = \{\theta : \|\theta\|_{a,b} \leq 1\}$ and $\Theta_q^* = \arg\max_{\theta \in \Theta} \mathbb{E}_{(x,y) \sim q} [g(\theta, x, y)]$. Similarly, let $\Omega = \{\omega : \|\omega\|_a \leq 1\}$ and $\Omega_q^* = \arg\max_{\omega \in \Omega} \mathbb{E}_{(x,y) \sim q} [\psi(\omega, x, y)]$. For the task of binary classification, if there exists $\{\theta^*, q^*\}$ satisfying Equation 1 and 2, then any $\hat{\theta} \in \arg\max_{\theta \in \Theta} \min_{(x,y) \in D} g(\theta, x, y)$ satisfies the following:*

- *$\hat{\theta}$ has directional support only on $\Omega_{q^*}^*$.*

- *For any $(x_1, y_1) \in spt(q^*)$, $f(\hat{\theta}, x_1, y_1) - f(\hat{\theta}, x_1, y_1') = \gamma^*$, where $y_1' \neq y_1$, i.e, all points in the support of $q^*$ are on the margin for any maximum margin solution.*

*Proof.* Let $\gamma^* = \max_{\theta \in \Theta} \min_{(x,y) \in D} g(\theta, x, y)$. Then, by Lemma 2, $\gamma^* = \mathbb{E}_{(x,y) \sim q^*} g(\theta^*, x, y)$.

Consider any $\hat{\theta} \in \arg\max_{\theta \in \Theta} \min_{(x,y) \in D} g(\theta, x, y)$. This means, that $\min_{(x,y) \in D} g(\hat{\theta}, x, y) = \gamma^*$. This implies that $\mathbb{E}_{(x,y) \sim q^*} g(\hat{\theta}, x, y) \geq \gamma^*$. However, by Equation 2, $\max_{\theta \in \Theta} \mathbb{E}_{(x,y) \sim q^*} g(\theta, x, y) = \gamma^*$. This implies that $\mathbb{E}_{(x,y) \sim q^*} g(\hat{\theta}, x, y) = \gamma^*$. Thus, $\hat{\theta}$ is also a maximizer of $\mathbb{E}_{(x,y) \sim q^*} g(\theta, x, y)$, and thus by Lemma 3, it only has directional support on $\Omega_{q^*}^*$.

Moreover, as $\mathbb{E}_{(x,y) \sim q^*} g(\hat{\theta}, x, y) = \gamma^*$, thus, for any $(x_1, y_1) \in spt(q^*)$, $f(\hat{\theta}, x_1, y_1) - f(\hat{\theta}, x_1, y_1') = \gamma^*$, where $y_1' \neq y_1$. $\qquad\square$

### F.2 MULTI-CLASS CLASSIFICATION

We will first provide the proof of Lemma 8.

**Lemma.** *Let $\Theta = \{\theta : \|\theta\|_{a,b} \leq 1\}$ and $\Theta_q'^* = \arg\max_{\theta \in \Theta} \mathbb{E}_{(x,y) \sim q} [g'(\theta, x, y)]$. Similarly, let $\Omega = \{\omega : \|\omega\|_a \leq 1\}$ and $\Omega_q'^* = \arg\max_{\omega \in \Omega} \mathbb{E}_{(x,y) \sim q} [\psi'(\omega, x, y)]$. Then:*

- ***Single neuron optimization**: Any $\theta \in \Theta_q'^*$ has directional support only on $\Omega_q'^*$.*

- ***Using multiple neurons**: If $b = \nu$ and $\omega_1^*, ..., \omega_m^* \in \Omega_q'^*$, then $\theta = \{\lambda_i \omega_i^*\}_{i=1}^m$ with $\sum \lambda_i^\nu = 1, \lambda_i \geq 0$ belongs to $\Theta_q'^*$.*

*Proof.* The proof follows the same strategy as the proof of Lemma 3 (Section F.1), following the linearity of $g'$. $\qquad\square$

Now, for ease of the reader, we will first restate Equation 3 and condition **C.1**.

$$\theta^* \in \arg\max_{\theta \in \Theta} \mathbb{E}_{(x,y) \sim q^*} [g'(\theta, x, y)].$$

**C.1** For any $(x, y) \in spt(q^*)$, it holds that $g'(\theta^*, x, y) = g(\theta^*, x, y)$. This translates to any label with non-zero weight being one of the incorrect labels where $f$ is maximized: $\{\ell \in \mathcal{Y} \setminus \{y\} : \tau(x, y)[\ell] > 0\} \subseteq \arg\max_{\ell \in \mathcal{Y} \setminus \{y\}} f(\theta^*, x)[\ell]$.

We will now the provide the proof of Lemma 9.

**Lemma.** *Let $\Theta = \{\theta : \|\theta\|_{a,b} \leq 1\}$ and $\Theta_q'^* = \arg\max_{\theta \in \Theta} \mathbb{E}_{(x,y) \sim q} [g'(\theta, x, y)]$. Similarly, let $\Omega = \{\omega : \|\omega\|_a \leq 1\}$ and $\Omega_q'^* = \arg\max_{\omega \in \Omega} \mathbb{E}_{(x,y) \sim q} [\psi'(\omega, x, y)]$. If $\exists \{\theta^*, q^*\}$ satisfying Equations 1 and 3, and **C.1** holds, then:*

- *$\theta^* \in \arg\max_{\theta \in \Theta} g(\theta, x, y)$*

- *Any $\hat{\theta} \in \arg\max_{\theta \in \Theta} \min_{(x,y) \in D} g(\theta, x, y)$ satisfies the following:*

  - *$\hat{\theta}$ has directional support only on $\Omega_{q^*}'^*$.*

– *For any $(x_1, y_1) \in spt(q^*)$, $f(\hat{\theta}, x_1, y_1) - \max_{y' \in \mathcal{Y} \backslash \{y_1\}} f(\hat{\theta}, x_1, y_1') = \gamma^*$, i.e, all points in the support of $q^*$ are on the margin for any maximum margin solution.*

*Proof.* For the first part, we will show that $\{\theta^*, q^*\}$ satisfy Equations 1 and 2, and then it follows from Lemma 2. As we have already assumed these satisfy Equation 1, we will show that they satisfy Equation 2.

Note that $g'(\theta, x, y) \geq g(\theta, x, y)$. Thus,

$$
\begin{aligned}
\mathbb{E}_{(x,y)\sim q^*}[g(\theta^*, x, y)] &\leq \max_{\theta \in \Theta} \mathbb{E}_{(x,y)\sim q^*}[g(\theta, x, y)] \\
&\leq \max_{\theta \in \Theta} \mathbb{E}_{(x,y)\sim q^*}[g'(\theta, x, y)] \\
&= \mathbb{E}_{(x,y)\sim q^*}[g'(\theta^*, x, y)]
\end{aligned}
$$

where the second inequality follows as $g' \geq g$ and the last equality follows as $\theta^*$ satisfies Equation 3. Now, as the pair also satisfies **C.1**, therefore $\mathbb{E}_{(x,y)\sim q^*}[g(\theta^*, x, y)] = \mathbb{E}_{(x,y)\sim q^*}[g'(\theta^*, x, y)]$. This means, that all inequalities in the above chain must be equality. Thus, $\theta^* \in \arg\max_{\theta \in \Theta} \mathbb{E}_{(x,y)\sim q^*}[g(\theta, x, y)]$. Thus, the pair $\{\theta^*, q^*\}$ satisfies Equation 1 and 2, and thus by Lemma 2, $\theta^* \in \arg\max_{\theta \in \Theta} g(\theta, x, y)$.

Let $\gamma^* = \max_{\theta \in \Theta} \min_{(x,y) \in D} g(\theta, x, y)$. Then, $\gamma^* = \mathbb{E}_{(x,y)\sim q^*} g(\theta^*, x, y)$. Consider any $\hat{\theta} \in \arg\max_{\theta \in \Theta} \min_{(x,y) \in D} g(\theta, x, y)$. This means, that $\min_{(x,y) \in D} g(\hat{\theta}, x, y) = \gamma^*$. This implies that $\mathbb{E}_{(x,y)\sim q^*} g(\hat{\theta}, x, y) \geq \gamma^*$. Since $g' \geq g$, it then folllows that $\mathbb{E}_{(x,y)\sim q^*} g'(\hat{\theta}, x, y) \geq \gamma^*$.

However, by Equation 3 and **C.1**, $\max_{\theta \in \Theta} \mathbb{E}_{(x,y)\sim q^*} g'(\theta, x, y) = \mathbb{E}_{(x,y)\sim q^*} g'(\theta^*, x, y) = \mathbb{E}_{(x,y)\sim q^*} g(\theta^*, x, y) = \gamma^*$. This implies that $\mathbb{E}_{(x,y)\sim q^*} g'(\hat{\theta}, x, y) = \gamma^*$. Thus, $\hat{\theta}$ is also a maximizer of $\mathbb{E}_{(x,y)\sim q^*} g'(\theta, x, y)$, and thus by Lemma 8, it only has directional support on $\Omega'^*_{q^*}$.

Moreover, as $\min_{(x,y) \in D} g(\hat{\theta}, x, y) = \gamma^*$, therefore, $\mathbb{E}_{(x,y)\sim q^*} g(\hat{\theta}, x, y) \geq \gamma^*$. However, as $g' \geq g$, therefore, $\mathbb{E}_{(x,y)\sim q^*} g(\hat{\theta}, x, y) \leq \mathbb{E}_{(x,y)\sim q^*} g'(\hat{\theta}, x, y) = \gamma^*$, as shown above. Thus, $\mathbb{E}_{(x,y)\sim q^*} g(\hat{\theta}, x, y) = \gamma^*$. Thus, we have $f(\hat{\theta}, x_1, y_1) - \max_{y' \in \mathcal{Y} \backslash \{y_1\}} f(\hat{\theta}, x_1, y_1') = g(\hat{\theta}, x_1, y_1) = \gamma^*$ for any $(x_1, y_1) \in \text{spt}(q^*)$. $\qquad\square$

# G  PROOFS FOR CYCLIC GROUPS(THEOREM 5)

As outlined in the blueprint in Section E.1, we will consider $q^*$ to be uniform on the dataset. We will use the uniform class weighting: $\tau(a, b)[c'] := 1/(p - 1)$ for all $c' \neq a + b$.

## G.1  PROOF THAT EQUATION 3 IS SATISFIED

*Proof.* Let

$$
\eta_{u,v,w}(\delta) := \mathbb{E}_{a,b}\left[(u(a) + v(b))^2 w(a + b - \delta)\right].
$$

We wish to find the solution to the following mean margin maximization problem:

$$
\arg\max_{u,v,w:\|u\|^2+\|v\|^2+\|w\|^2 \leq 1} (\eta_{u,v,w}(0) - \mathbb{E}_{\delta \neq 0}[\eta_{u,v,w}(\delta)]) = \frac{p}{p-1}(\eta_{u,v,w}(0) - \mathbb{E}_\delta[\eta_{u,v,w}(\delta)]).
$$

(4)

First, note that $\mathbb{E}_c[w(c)] = 0$, because shifting the mean of $w$ does not affect the margin. It follows that

$$
\mathbb{E}_{a,b}\left[(u(a)^2 w(a + b - \delta)\right] = \mathbb{E}_a\left[u(a)^2 \mathbb{E}_b[w(a + b - \delta)]\right] = \mathbb{E}_a\left[u(a)^2 \mathbb{E}_b[w(b)]\right] = 0,
$$

and similarly for the $v(b)^2$ component of $\eta$, so we can rewrite (4) as

$$
\arg\max_{u,v,w:\|u\|^2+\|v\|^2+\|w\|^2 \leq 1} \frac{2p}{p-1}(\tilde{\eta}_{u,v,w}(0) - \mathbb{E}_\delta[\tilde{\eta}_{u,v,w}(\delta)]),
$$

where

$$\tilde{\eta}_{u,v,w}(\delta) := \mathbb{E}_{a,b}\left[u(a)v(b)w(a+b-\delta)\right].$$

Let $\rho := e^{2\pi i/p}$, and let $\hat{u}, \hat{v}, \hat{w}$ be the discrete Fourier transforms of $u$, $v$, and $w$ respectively. Then we have:

$$\tilde{\eta}_{u,v,w}(\delta) = \mathbb{E}_{a,b}\left[\left(\frac{1}{p}\sum_{j=0}^{p-1}\hat{u}(j)\rho^{ja}\right)\left(\frac{1}{p}\sum_{k=0}^{p-1}\hat{v}(k)\rho^{kb}\right)\left(\frac{1}{p}\sum_{\ell=0}^{p-1}\hat{w}(\ell)\rho^{\ell(a+b-\delta)}\right)\right]$$

$$= \frac{1}{p^3}\sum_{j,k,\ell}\hat{u}(j)\hat{v}(k)\hat{w}(\ell)\rho^{-\ell\delta}\left(\mathbb{E}_a\rho^{(j+\ell)a}\right)\left(\mathbb{E}_b\rho^{(k+\ell)b}\right)$$

$$= \frac{1}{p^3}\sum_j\hat{u}(j)\hat{v}(j)\hat{w}(-j)\rho^{j\delta} \qquad \text{(only terms where } j+\ell = k+\ell = 0 \text{ survive)}$$

Hence, we need to maximize

$$\frac{2p}{p-1}(\tilde{\eta}_{u,v,w}(0) - \mathbb{E}_\delta\left[\tilde{\eta}_{u,v,w}(\delta)\right]) \tag{5}$$

$$= \frac{2p}{p-1}\left(\frac{1}{p^3}\sum_j\hat{u}(j)\hat{v}(j)\hat{w}(-j) - \frac{1}{p^3}\sum_j\hat{u}(j)\hat{v}(j)\hat{w}(-j)(\mathbb{E}_\delta\rho^{j\delta})\right)$$

$$= \frac{2}{(p-1)p^2}\sum_{j\neq 0}\hat{u}(j)\hat{v}(j)\hat{w}(-j). \tag{6}$$

We have arrived at the crux of why any max margin solution must be sparse in the Fourier domain: in order to maximize expression 6, we must concentrate the mass of $\hat{u}$, $\hat{v}$, and $\hat{w}$ on the same frequencies, the fewer the better. We will now work this out carefully. Since $u, v, w$ are real-valued, we have

$$\hat{u}(-j) = \overline{\hat{u}(j)}, \hat{v}(-j) = \overline{\hat{v}(j)}, \hat{w}(-j) = \overline{\hat{w}(j)}$$

for all $j \in \mathbb{Z}_p$. Let $\theta_u, \theta_v, \theta_w \in [0, 2\pi)^p$ be the phase components of $u, v, w$ respectively; so, e.g., for $\hat{u}$:

$$\hat{u}(j) = |\hat{u}(j)|\exp(i\theta_u(j)).$$

Then, for odd $p$, expression 6 becomes:

$$\frac{2}{(p-1)p^2}\sum_{j=1}^{(p-1)/2}\left[\hat{u}(j)\hat{v}(j)\overline{\hat{w}(j)} + \overline{\hat{u}(j)\hat{v}(j)}\hat{w}(j)\right]$$

$$= \frac{2}{(p-1)p^2}\sum_{j=1}^{(p-1)/2}|\hat{u}(j)||\hat{v}(j)||\hat{w}(j)|\left[\exp(i(\theta_u(j)+\theta_v(j)-\theta_w(j))) + \exp(i(-\theta_u(j)-\theta_v(j)+\theta_w(j)))\right]$$

$$= \frac{4}{(p-1)p^2}\sum_{j=1}^{(p-1)/2}|\hat{u}(j)||\hat{v}(j)||\hat{w}(j)|\cos(\theta_u(j)+\theta_v(j)-\theta_w(j)).$$

Thus, we need to optimize:

$$\max_{u,v,w:\|u\|^2+\|v\|^2+\|w\|^2\leq 1}\frac{4}{(p-1)p^2}\sum_{j=1}^{(p-1)/2}|\hat{u}(j)||\hat{v}(j)||\hat{w}(j)|\cos(\theta_u(j)+\theta_v(j)-\theta_w(j)). \tag{7}$$

By Plancherel's theorem, the norm constraint is equivalent to

$$\|\hat{u}\|^2 + \|\hat{v}\|^2 + \|\hat{w}\|^2 \leq p,$$

so the choice of $\theta_u(j), \theta_v(j), \theta_w(j)$ is unconstrained. Therefore, we can (and must) choose them to satisfy $\theta_u(j) + \theta_v(j) = \theta_w(j)$, so that $\cos(\theta_u(j) + \theta_v(j) - \theta_w(j)) = 1$ is maximized for each $j$

(unless the amplitude part of the $j$th term is 0, in which case the phase doesn't matter). The problem is thus further reduced to:

$$\max_{|\hat{u}|,|\hat{v}|,|\hat{w}|:\|\hat{u}\|^2+\|\hat{v}\|^2+\|\hat{w}\|^2\leq p} \frac{4}{(p-1)p^2} \sum_{j=1}^{(p-1)/2} |\hat{u}(j)||\hat{v}(j)||\hat{w}(j)|. \tag{8}$$

By the inequality of quadratic and geometric means,

$$|\hat{u}(j)||\hat{v}(j)||\hat{w}(j)| \leq \left(\frac{|\hat{u}(j)|^2 + |\hat{v}(j)|^2 + |\hat{w}(j)|^2}{3}\right)^{3/2}. \tag{9}$$

Let $z : \{1, \ldots, \frac{p-1}{2}\} \to \mathbb{R}$ be defined as $z(j) := |\hat{u}(j)|^2 + |\hat{v}(j)|^2 + |\hat{w}(j)|^2$. Then, since we must have $\hat{u}(0) = \hat{v}(0) = \hat{w}(0) = 0$ in the optimization above, we can upper-bound expression 8 by

$$\frac{4}{(p-1)p^2} \cdot \max_{\|z\|_1\leq\frac{p}{2}} \sum_{j=1}^{(p-1)/2} \left(\frac{z(j)}{3}\right)^{3/2}$$

$$\leq \frac{4}{3^{3/2}(p-1)p^2} \cdot \max_{\|z\|_1\leq\frac{p}{2}} \left(\sum_{j=1}^{(p-1)/2} z(j)^2\right)^{1/2} \cdot \left(\sum_{j=1}^{(p-1)/2} z(j)\right)^{1/2} \quad \text{(Cauchy-Schwartz)}$$

$$= \frac{2^{3/2}}{3^{3/2}(p-1)p^{3/2}} \cdot \max_{\|z\|_1\leq\frac{p}{2}} \|z\|_2$$

$$\leq \frac{2^{3/2}}{3^{3/2}(p-1)p^{3/2}} \cdot \frac{p}{2} = \sqrt{\frac{2}{27}} \cdot \frac{1}{p^{1/2}(p-1)}.$$

The only way to turn inequality 9 into an equality is to set $|\hat{u}(j)| = |\hat{v}(j)| = |\hat{w}(j)|$, and the only way to achieve $\|z\|_2 = \frac{p}{2}$ is to place all the mass on a single frequency, so the only possible way to achieve the upper bound is to set

$$|\hat{u}(j)| = |\hat{v}(j)| = |\hat{w}(j)| = \begin{cases} \sqrt{p/6} & \text{if } j = \pm\zeta \\ 0 & \text{otherwise} \end{cases}.$$

for some frequency $\zeta \in \{1, \ldots, \frac{p-1}{2}\}$. In this case, we indeed match the upper bound:

$$\frac{4}{(p-1)p^2} \cdot \left(\frac{p}{6}\right)^{3/2} = \sqrt{\frac{2}{27}} \cdot \frac{1}{p^{1/2}(p-1)}.$$

so this is the maximum margin.

Putting it all together, and abusing notation by letting $\theta_u^* := \theta_u(\zeta)$, we obtain that all neurons maximizing the expected class-weighted margin are of the form (up to scaling):

$$u(a) = \frac{1}{p}\sum_{j=0}^{p-1} \hat{u}(j)\rho^{ja}$$

$$= \frac{1}{p}\left[\hat{u}(\zeta)\rho^{\zeta a} + \hat{u}(-\zeta)\rho^{-\zeta a}\right]$$

$$= \frac{1}{p}\left[\sqrt{\frac{p}{6}}\exp(i\theta_u^*)\rho^{\zeta a} + \sqrt{\frac{p}{6}}\exp(-i\theta_u^*)\rho^{-\zeta a}\right]$$

$$= \sqrt{\frac{2}{3p}}\cos(\theta_u^* + 2\pi\zeta a/p)$$

and

$$v(b) = \sqrt{\frac{2}{3p}}\cos(\theta_v^* + 2\pi\zeta b/p)$$

$$w(c) = \sqrt{\frac{2}{3p}}\cos(\theta_w^* + 2\pi\zeta c/p)$$

for some phase offsets $\theta_u^*, \theta_v^*, \theta_w^* \in \mathbb{R}$ satisfying $\theta_u^* + \theta_v^* = \theta_w^*$ and some $\zeta \in \mathbb{Z}_p \setminus \{0\}$ (where $\zeta$ is the same for $u$, $v$, and $w$). $\qquad\square$

It remains to construct a network $\theta^*$ which uses neurons of the above form and satisfies condition **C.1** and Equation 1 with respect to $q = \text{unif}(\mathbb{Z}_p)$.

### G.2    Proof that condition **C.1** and Equation 1 are satisfied

*Proof.* Our $\theta^*$ will consist of $4(p-1)$ neurons: 8 neurons for each of the frequencies $1, \ldots, \frac{p-1}{2}$. Consider a given frequency $\zeta$. For brevity, let $\cos_\zeta(x)$ denote $\cos(2\pi\zeta x/p)$, and similarly for $\sin_\zeta(x)$. First, we observe:

$$\begin{aligned}
\cos_\zeta(a+b-c) &= \cos_\zeta(a+b)\cos_\zeta(c) + \sin_\zeta(a+b)\sin_\zeta(c) \\
&= \cos_\zeta(a)\cos_\zeta(b)\cos_\zeta(c) - \sin_\zeta(a)\sin_\zeta(b)\cos_\zeta(c) \\
&\quad + \sin_\zeta(a)\cos_\zeta(b)\sin_\zeta(c) + \cos_\zeta(a)\sin_\zeta(b)\sin_\zeta(c)
\end{aligned}$$

Each of these four terms can be implemented by a pair of neurons $\phi_1, \phi_2$. Consider the first term, $\cos_\zeta(a)\cos_\zeta(b)\cos_\zeta(c)$. For the first neuron $\phi_1$, set $u_1(\cdot), v_1(\cdot), w_1(\cdot) := \cos_\zeta(\cdot)$, and for $\phi_2$, set $u_2(\cdot) := \cos_\zeta(\cdot)$ and $v_2(\cdot), w_2(\cdot) := -\cos_\zeta(\cdot)$. These can be implemented in the form we derived by setting $(\theta_u^*, \theta_v^*, \theta_w^*)$ to $(0, 0, 0)$ for the first neuron and $(0, \pi, \pi)$ for the second.

Adding these two neurons, we obtain:

$$\begin{aligned}
\phi_1(a,b) + \phi_2(a,b) &= (\cos_\zeta(a) + \cos_\zeta(a))^2 \cos_\zeta(c) + (\cos_\zeta(a) - \cos_\zeta(a))^2(-\cos_\zeta(c)) \\
&= 4\cos_\zeta(a)\cos_\zeta(b)\cos_\zeta(c)
\end{aligned}$$

Similarly, each of the other three terms can be implemented by pairs of neurons, by setting $(\theta_u^*, \theta_v^*, \theta_w^*)$ to

1. $(\frac{\pi}{2}, -\frac{\pi}{2}, 0)$ and $(\frac{\pi}{2}, \frac{\pi}{2}, \pi)$
2. $(-\frac{\pi}{2}, 0, -\frac{\pi}{2})$ and $(-\frac{\pi}{2}, \pi, \frac{\pi}{2})$
3. $(0, -\frac{\pi}{2}, -\frac{\pi}{2})$ and $(0, \frac{\pi}{2}, \frac{\pi}{2})$

If we include such a collection of 8 neurons for every frequency $\zeta \in \{1, \ldots, \frac{p-1}{2}\}$, the resulting network will compute the function

$$\begin{aligned}
f(a,b) &= \sum_{\zeta=1}^{(p-1)/2} \cos_\zeta(a+b-c) \\
&= \sum_{\zeta=1}^{p-1} \frac{1}{2} \cdot \exp(2\pi i \zeta(a+b-c)/p) \\
&= \begin{cases} \frac{p-1}{2} & \text{if } a+b=c \\ 0 & \text{otherwise} \end{cases}
\end{aligned}$$

The scaling constant $\lambda$ for each neuron can be chosen so that the network has $L_{2,3}$-norm 1. For this network, every datapoint is on the margin, so $q = \text{unif}(\mathbb{Z}_p)$ is trivially supported on points on the margin, satisfying Equation 1. And for each input $(a, b)$, $f$ takes the same value on all incorrect labels $c'$, satisfying **C.1**. $\square$

### G.3    Proof that all frequencies are used

*Proof.* For this proof, we need to introduce the multidimensional discrete Fourier transform. For a function $f : \mathbb{Z}_p^3 \to \mathbb{C}$, the multidimensional DFT of $f$ is defined as:

$$\hat{f}(j,k,\ell) := \sum_{a \in \mathbb{Z}_p} e^{-2\pi i \cdot ja/p} \left( \sum_{b \in \mathbb{Z}_p} e^{-2\pi i \cdot jb/p} \left( \sum_{c \in \mathbb{Z}_p} e^{-2\pi i \cdot jc/p} f(a,b,c) \right) \right)$$

for all $j, k, \ell \in \mathbb{Z}$.

To simplify the notation, let $\theta_u = \theta_u^* \cdot \frac{p}{2\pi}$, so

$$u(a) = \sqrt{\frac{2}{3p}} \cos_p(\theta_u + \zeta a).$$

Let

$$
\begin{aligned}
f(a, b, c) &= \sum_{h=1}^{H} \phi_h(a, b, c) \\
&= \sum_{h=1}^{H} (u_h(a) + v_h(b))^2 \, w_h(c) \\
&= \left(\frac{2}{3p}\right)^{3/2} \sum_{h=1}^{H} (\cos_p(\theta_{u_h} + \zeta_h a) + \cos_p(\theta_{v_h} + \zeta_h b))^2 \cos_p(\theta_{w_h} + \zeta_h c)
\end{aligned}
$$

be the function computed by an arbitrary margin-maximizing network of width $H$, where each neuron is of the form derived earlier.

Each neuron $\phi$ can be split into three terms:

$$\phi(a, b, c) = \phi^{(1)}(a, b, c) + \phi^{(2)}(a, b, c) + \phi^{(3)}(a, b, c) := u(a)^2 w(c) + v(b)^2 w(c) + 2u(a)v(b)w(c)$$

$\widehat{\phi^{(1)}}(j, k, \ell)$ is nonzero only for $k = 0$, and $\widehat{\phi^{(2)}}(j, k, \ell)$ is nonzero only for $j = 0$. For the third term, we have

$$\widehat{\phi^{(3)}}(j, k, \ell) = 2 \sum_{a,b,c \in \mathbb{Z}_p} u(a)v(b)w(c)\rho^{-(ja+kb+\ell c)} = 2\hat{u}(j)\hat{v}(k)\hat{w}(\ell).$$

In particular,

$$
\begin{aligned}
\hat{u}(j) &= \sum_{a \in \mathbb{Z}_p} \sqrt{\frac{2}{3p}} \cos_p(\theta_u + \zeta a)\rho^{-ja} \\
&= (6p)^{-1/2} \sum_{a \in \mathbb{Z}_p} \left(\rho^{\theta_u + \zeta a} + \rho^{-(\theta_u + \zeta a)}\right) \rho^{-ja} \\
&= (6p)^{-1/2} \left(\rho^{\theta_u} \sum_{a \in \mathbb{Z}_p} \rho^{(\zeta - j)a} + \rho^{-\theta_u} \sum_{a \in \mathbb{Z}_p} \rho^{-(\zeta + j)a}\right) \\
&= \begin{cases} \sqrt{p/6} \cdot \rho^{\theta_u} & \text{if } j = \zeta \\ \sqrt{p/6} \cdot \rho^{-\theta_u} & \text{if } j = -\zeta \\ 0 & \text{otherwise} \end{cases}
\end{aligned}
$$

and similarly for $\hat{v}$ and $\hat{w}$. $\zeta$ was defined to be nonzero, so the $\zeta = 0$ case is ignored. Thus, $\widehat{\phi^{(3)}}(j, k, \ell)$ is nonzero only when $j, k, \ell$ are all $\pm\zeta$. We can conclude that $\hat{\phi}(j, k, \ell)$ can only be nonzero if one of the following conditions holds:

1. $j = 0$

2. $k = 0$

3. $j, k, \ell = \pm\zeta$.

Independent of the above considerations, we know by Lemma 9 that the function $f$ implemented by the network has equal margin across different inputs and across different classes for the same input. In other words, $f$ can be decomposed as

$$f(a, b, c) = f_1(a, b, c) + f_2(a, b, c)$$

where

$$f_1(a, b, c) = F(a, b)$$

for some $F : \mathbb{Z}_p \times \mathbb{Z}_p \to \mathbb{R}$, and

$$f_2(a, b, c) = \lambda \cdot \mathbf{1}_{a+b=c}$$

where $\lambda > 0$ is the margin of $f$.

The Fourier transforms of $f_1$ and $f_2$ are

$$\hat{f}_1(j, k, l) = \begin{cases} \hat{F}(j, k) & \text{if } \ell = 0 \\ 0 & \text{otherwise} \end{cases}$$

and

$$\hat{f}_2(j, k, l) = \begin{cases} \lambda p^2 & \text{if } j = k = -\ell \\ 0 & \text{otherwise} \end{cases}.$$

Hence, when $j = k = -\ell \neq 0$, we must have $\hat{f}(j, k, \ell) > 0$. But then, from the conditions under which each neuron's DFT $\hat{\phi}$ is nonzero, it must follow that there is at least one neuron for each frequency. $\qquad\square$

## H   PROOFS FOR SPARSE PARITY

**Theorem.** *Consider a single hidden layer neural network of width $m$ with the activation function given by $x^k$, i.e, $f(x) = \sum_{i=1}^{m} (u_i^\top x)^k w_i$, where $u_i \in \mathbb{R}^n$ and $w_i \in \mathbb{R}^2$, trained on the $(n, k)-$sparse parity task. Without loss of generality, assume that the first coordinate of $w_i$ corresponds to the output for class $y = +1$. Denote the vector $[1, -1]$ by $\mathbf{b}$. Provided $m \geq 2^{k-1}$, the $L_{2,k+1}$ maximum margin is:*

$$k!\sqrt{2(k+1)^{-(k+1)}}.$$

*Any network achieving this margin satisfies the following conditions:*

1. *For every $i$ having $\|u_i\| > 0$, $spt(u_i) = S$, $w_i$ lies in the span of $\mathbf{b}$ and $\forall j \in S$, $|u_i[j]| = \|w_i\|$.*

2. *For every $i$, $\left(\Pi_{j \in S} u_i[j]\right)\left(w_i^\top \mathbf{b}\right) \geq 0$.*

*Proof.* We will consider $q^*$ to be equally distributed on the dataset and optimize the class-weighted margin as defined in Equation 3. We will consider the weight $\tau(x, y)[y'] = 1$ for $y' \neq y$. Also, let $\mathbf{a}$ denote the vector $[1, 1]$ and $\mathbf{b}$ denote the vector $[1, -1]$. Then, any $w_i \in \mathbb{R}^2$ can be written as $w_i = \frac{1}{\sqrt{2}} [\alpha_i \mathbf{a} + \beta_i \mathbf{b}]$ for some $\alpha_i, \beta_i \in \mathbb{R}$.

First, using lemma 8, we can say that one neuron maximizers of class-weighted margin are given by

$$\underset{\|[u,w]\|_2 \leq 1}{\arg\max} \; \mathbb{E}_{(x,y) \sim D} \left[\phi(\{u, w\}, x)[y] - \phi(\{u, w\}, x)[y']\right]$$

where $y' = -y$, $\phi(\{u, w\}, x) = (u^\top x)^k w$ and $\|[u, w]\|_2$ represents the 2-norm of the concatenation of $u$ and $w$.

Considering that $y \in \{\pm 1\}$ and $w = \frac{1}{\sqrt{2}} [\alpha \mathbf{a} + \beta \mathbf{b}]$, we can say $\phi(\{u, w\}, x)[y] = \frac{1}{\sqrt{2}} (u^\top x)^k [\alpha + y\beta]$. Thus, we can say

$$\begin{aligned}
\mathbb{E}_{(x,y) \sim D} \left[\phi(\{u, w\}, x)[y] - \phi(\{u, w\}, x)[y']\right] &= \sqrt{2} \mathbb{E}_{(x,y) \sim D} \left[(u^\top x)^k \beta y\right] \\
&= \sqrt{2} \mathbb{E}_{(x,y) \sim D} \left[(u^\top x)^k \beta \Pi_{i \in S} x_i\right] \\
&= \sqrt{2} k! \left(\Pi_{i \in S} u_i\right) \beta
\end{aligned}$$

where in the last step, all other terms are zero by symmetry of the dataset.

Clearly, under the constraint $\|u\|^2 + \alpha^2 + \beta^2 \leq 1$ (where $\|w\|^2 = \alpha^2 + \beta^2$), this is maximized when $u_i = 0$ for $i \notin S$, $\alpha = 0$, $u_i = \pm\frac{1}{\sqrt{k+1}}$ and $\beta = \pm\frac{1}{\sqrt{k+1}}$, with $(\Pi_{i \in S} u_i)\beta > 0$.

Now, using Lemma 8, we will create a network using these optimal neurons such that it satisfies **C.1**, and Equations 1 and 3, thus concluding by Lemma 9. **C.1** holds trivially as this is a binary classification task, so $g' = g$.

Consider a maximal subset $A \subset \{\pm1\}^k$ such that if $\sigma \in A$, then $-\sigma \notin A$ and for any $\sigma \in A, \sigma_1 = 1$. Now, consider a neural network having $2^{k-1}$ neurons given by

$$f(\theta, x) = \frac{1}{2^{k-1}} \sum_{\sigma \in A} \left(\sum_{i=1}^{k} \frac{\sigma_i}{\sqrt{k+1}} x_{S_i}\right)^k \frac{\left(\Pi_{i=1}^k \sigma_i\right)}{\sqrt{k+1}} \frac{1}{\sqrt{2}} \boldsymbol{b} = \frac{1}{\sqrt{2}} k! (k+1)^{-(k+1)/2} \left(\Pi_{i \in S} x_i\right) \boldsymbol{b}$$

By Lemma 8, the above neural network also maximizes the class-weighted mean margin. Moreover, it also satisfies Equation 1, as every term other than $\Pi x_{S_i}$ cancels out in the sum.

Consider any monomial $T$ which depends only on $S' \subset S$. Consider any one of the terms in $f(x)$ and let the coefficient of $T$ in the term given by $c_T$. Consider another term in $f(x)$, where, for some $i \in S \setminus S'$ and $j = k+1$, $\sigma_i$ and $\sigma_j$ are flipped. For this term, the coefficient of $T$ will be $-c_T$, as for all $i \in S'$, $\sigma_i$ is the same, but $\sigma_{k+1}$ is different. Thus, for any such monomial, its coefficient in expanded $f(x)$ will be $0$ as terms will always exist in these pairs.

Thus, $f(\theta, x)$ satisfies **C.1**, Equation 1 and 3, hence, by Lemma 9, any maximum margin solution satisfies the properties stated in Theorem 6. $\qquad\square$

# I  ADDITIONAL GROUP REPRESENTATION THEORY PRELIMINARIES

In this section we properly define relevant results from group representation theory used in the proof of Theorem 7. We also refer the reader to Kosmann-Schwarzbach et al. (2010), one of many good references for representation theory.

**Definition 1.** *A linear representation of a group $G$ is a finite dimensional complex vector space $V$ and a group homomorphism $R : G \to GL(V)$. We denote such a representation by $(R, V)$ or simply just $R$. The dimension of the representation $R$, denoted $d_R$, equals the dimension of the vector space $V$.*

In our case we are only concerned with finite groups with real representations, i.e. $V = \mathbb{R}^d$ and each representation $R$ maps group elements to real invertible $d \times d$ matrices. Furthermore, we are only concerned with *unitary* representations $R$, i.e. $R(g)$ is unitary for every $g$. It is a known fact that every representation of a finite group can be made unitary, in the following sense:

**Theorem 10** (Kosmann-Schwarzbach et al. (2010), Theorem 1.5.). *Every representation of a finite group $(R, V)$ is unitarizable, i.e. there is a scalar product on $V$ such that $R$ is unitary.*

Also of particular interest are irreducible representations.

**Definition 2.** *A representation $(R, V)$ of $G$ is irreducible if $V \neq \{0\}$ and the only vector subspaces of $V$ invariant under $R$ are $\{0\}$ or $V$ itself.*

A well-known result is Maschke's Theorem, which states that every finite-dimensional representation of a finite group is completely reducible; thus it suffices to consider a fundamental set of irreducible unitary representations in our analysis.

**Theorem 11** (Maschke's Theorem.). *Every finite-dimensional representation of a finite group is a direct sum of irreducible representations.*

**Theorem 12** (Kosmann-Schwarzbach et al. (2010), Theorem 3.4.). *Let $G$ be a finite group. If $R_1, ..., R_K$ denote the irreducible representations of $G$, then $|G| = \sum_{n=1}^{K} d_{R_n}^2$, where $d_{R_n}$ represents the dimensionality of $R_n$.*

The theory about characters of representations and orthogonality relations are essential for our max margin analysis. This is a rich area of results, and we only list those that are directly used in our proofs.

**Definition 3.** *Let $(R, V)$ be a representation of $G$. the character of $R$ is the function $\chi_R : G \to \mathbb{R}$ defined as $\chi_R(g) = \text{tr}(R(g))$ for each $g \in G$.*

For each conjugacy class of $G$, the character of $R$ is constant (this can easily be verified via properties of the matrix trace). More generally, functions which are constant for each conjugacy class are called *class functions* on $G$. Given the characters across inequivalent irreducible representations, one can construct a "*character table*" for a group $G$ in which the columns correspond to the conjugacy classes of a group, and whose rows correspond to inequivalent irreducible representations of a group. The entries of the character table correspond to the character for the representation at that given row, evaluated on the conjugacy class at that given column.

Characters of inequivalent irreducible representations are in fact orthogonal, which follow from the orthogonality relations of representation matrix elements. For a unitary irreducible representation $R$, define the vector $R_{(i,j)} = (R(g)_{(i,j)})_{g \in G}$ with entries being the $(i, j)$th entry of the matrix output for each $g \in G$ under $R$. We have the following result.

**Proposition 1** (Kosmann-Schwarzbach et al. (2010), Corollary 2.10.). *Let $(R_1, V_1)$ and $(R_2, V_2)$ be unitary irreducible representations of $G$. Choosing two orthonormal bases in $V_1$ and $V_2$, the following holds:*

1. *If $R_1$ and $R_2$ are inequivalent, then for every $i, j, k, l$, we have $\langle R_{1(i,j)}, R_{2(k,l)} \rangle = 0$.*

2. *If $R_1 = R_2 = R$ and $V_1 = V_2 = V$, then for every $i, j, k, l$, we have $\langle R_{(i,j)}, R_{(k,l)} \rangle = \frac{1}{d_R} \delta_{ik} \delta_{jl}$, where $\delta_{ik} = \mathbb{1}[i = k]$.*

**Theorem 13** (Kosmann-Schwarzbach et al. (2010), Theorem 2.11.). *Let $G$ be a finite group. If $R_1$ and $R_2$ are inequivalent irreducible representations of $G$, then $\langle \chi_{R_1}, \chi_{R_2} \rangle = 0$. If $R$ is an irreducible representation of $G$, then $\langle \chi_R, \chi_R \rangle = 1$.*

A fundamental result about characters is that the irreducible characters of $G$ form an orthonormal set in $L^2(G)$ (Kosmann-Schwarzbach et al. (2010), Theorem 2.12.). This implies the following result, which states that the irreducible characters form an orthonormal basis in the vector space of class functions on $G$ taking values in $\mathbb{R}$. Since this vector space has dimension equal to the number of conjugacy classes of $G$, it also follows that the number of equivalence classes of irreducible representations is the number of conjugacy classes. In other words, the character table is square for every finite group.

**Theorem 14** (Kosmann-Schwarzbach et al. (2010), Theorem 3.6.). *The irreducible characters form an orthonormal basis of the vector space of character functions.*

In section J of the Appendix, we rigorously define the basis vectors for network weights based on the representation matrix elements defined in Proposition 1, and establish the properties they satisfy, which are key to our analysis.

### I.1 A Concrete Example: Symmetric Group

The symmetric group $S_n$ consists of the permutations over a set of cardinality $n$. The order of the group is $n!$. It is a fact that every permutation can be written as a product of *transpositions*— a permutation which swaps two elements. We can associate with each permutation the parity of the number of transpositions needed, which is independent of the choice of decomposition.

We will provide a concrete description of the representation theory for $S_5$, which is a central group of study in this paper. It has 7 conjugacy classes, which we denote as $\{e, (1\ 2), (1\ 2)(3\ 4), (1\ 2\ 3), (1\ 2\ 3\ 4), (1\ 2\ 3\ 4\ 5), (1\ 2)(3\ 4\ 5)\}$ (selecting one representative from each conjugacy class). It also has 7 irreducible representations. Apart from the trivial representation, it has another 1-dimensional *sign* representation representing the parity of a permutation.

The symmetric group also has an $n$-dimensional representation which is the *natural permutation representation*, mapping permutations to *permutation matrices* which shuffle the $n$ coordinates. It turns out that this is in fact reducible, since this has the trivial subrepresentation consisting of vectors whose coordinates are all equal. Decomposing this representation into irreducible representations results in the trivial representation and what is called the *standard* representation of dimension $n -$

| class | $e$ | (1 2) | (1 2)(3 4) | (1 2 3) | (1 2 3 4) | (1 2 3 4 5) | (1 2)(3 4 5) |
|---|---|---|---|---|---|---|---|
| size | 1 | 10 | 15 | 20 | 30 | 24 | 20 |
| $R_1$ | 1 | 1 | 1 | 1 | 1 | 1 | 1 |
| $R_2$(sign) | 1 | $-1$ | 1 | 1 | $-1$ | 1 | $-1$ |
| $R_3$(standard) | 4 | $-2$ | 0 | 1 | 0 | $-1$ | 1 |
| $R_4$(standard $\otimes$ sign) | 4 | 2 | 0 | 1 | 0 | $-1$ | $-1$ |
| $R_5$(5d_a) | 5 | 1 | 1 | $-1$ | $-1$ | 0 | 1 |
| $R_6$(5d_b) | 5 | $-1$ | 1 | $-1$ | 1 | 0 | $-1$ |
| $R_7$(6d) | 6 | 0 | $-2$ | 0 | 0 | 1 | 0 |

Table 1: Character table of $S_5$.

1. It has another $n-1$-dimensional representation, which is the *product of sign and standard* representations.

The final three representations of $S_5$ are higher-dimensional, with dimensions $5, 5$, and $6$. We denote them as 5d_a, 5d_b, and 6d. We give the character table of $S_5$ in Table 1, which will be useful for calculating the value of the max margin which we theoretically derive.

## J  PROOFS FOR FINITE GROUPS WITH REAL REPRESENTATIONS

In this section we prove that for finite groups with real representations, all max margin solutions have neurons which only use a single irreducible representation.

**Theorem.** *Consider a single hidden layer neural network of width $m$ with quadratic activation trained on learning group composition for $G$ with real irreducible representations. Provided $m \geq 2\sum_{n=2}^{K} d_{R_n}^{3}$ and $\sum_{n=2}^{K} d_{R_n}^{1.5} \chi_{R_n}(C) < 0$ for every non-trivial conjugacy class $C$, the $L_{2,3}$ maximum margin is:*

$$\gamma^* = \frac{2}{3\sqrt{3|G|}} \frac{1}{\left(\sum_{n=2}^{K} d_{R_n}^{2.5}\right)}.$$

*Any network achieving this margin satisfies the following conditions:*

1. *For every neuron, there exists a non-trivial representation such that the input and output weight vectors are spanned only by that representation.*

2. *There exists at least one neuron spanned by each representation (except for the trivial representation) in the network.*

Let $R_1, \ldots, R_K$ be the unitary irreducible representations and let $C_1, \ldots, C_K$ be the conjugacy classes of a finite group $G$ with real representations. We fix $R_1$ to be the trivial one-dimensional representation mapping $R_1(g) = 1$ for all $g \in G$ and $C_1$ to be the trivial conjugacy class $C_1 = \{e\}$. For each of these representations $(V, R)$ of the group $G$, where $R : G \to V$, we will consider the $|G|$-dimensional vectors by fixing one position in the matrix $R(g)$ for all $g \in G$, i.e. vectors $(R(g)_{i,j})_{g \in G}$ for some $i, j \in [d_V]$. These form a set of $|G|$ vectors which we will denote $\rho_1, ..., \rho_{|G|}$ ($\rho_1$ is always the vector corresponding to the trivial representation). These vectors in fact form an orthogonal basis, and satisfy additional properties established in the following lemma.

**Lemma 15.** *The set of vectors $\rho_1, ..., \rho_{|G|}$ satisfy the following properties:*

1. $\sum_{a \in G} \rho_i(a)\rho_j(a) = 0$ *for $i \neq j$. (Orthogonality)*

2. $\sum_{a \in G} \rho_i(a)^2 = |G|/d_V$ *for all $i$, where $d_V$ is the dimensionality of the vector space $V$ corresponding to the representation that $\rho_i$ belongs to.*

3. *For all the $\rho_j$ which correspond to off-diagonal entries of a representation, $\sum_{a \in C_i} \rho_j[a] = 0$, i.e, the sum of elements within the same conjugacy class is 0.*

4. *If $\rho_j$ and $\rho_k$ correspond to different diagonal entries within the same representation, then $\sum_{a \in C_i} \rho_j[a] = \sum_{a \in C_i} \rho_k[a]$, i.e, for the diagonal entries, the sum for a given conjugacy class is invariant with the position of the diagonal element.*

*Proof.* The first two properties are the orthogonality relations of unitary representation matrix elements (Proposition 1), and the last two points follow additionally from Proposition 2.7 and Proposition 2.8 in Kosmann-Schwarzbach et al. (2010). □

Since this set of $|G|$-dimensional vectors are orthogonal to each other, each set of weights for a neuron in our architecture can be expressed as a linear combination of these basis vectors

$$u = \sum_{i \in [|G|]} \alpha_i \rho_i, \quad v = \sum_{i \in [|G|]} \beta_i \rho_i, \quad w = \sum_{i \in [|G|]} \gamma_i \rho_i.$$

It will also be useful to define the matrices $\boldsymbol{\alpha}_{R_i}, \boldsymbol{\beta}_{R_i}, \boldsymbol{\gamma}_{R_i}$ for each irreducible representation $R_i$ of $G$ which consist of the coefficients for $u$, $v$, and $w$ corresponding to each entry in the representation matrix.

Let $h_{u,v,w}(c) := \mathbb{E}_{a,b}\left[(u(a) + v(b))^2 w(a \circ b \circ c)\right]$. Recall we seek solutions for the following *weighted* margin maximization problem

$$h_{u,v,w}(e) - \sum_{c \neq e} \tau_c h_{u,v,w}(c), \text{ where } \sum_{c \neq e} \tau_c = 1. \tag{10}$$

Note that if we substitute the weights $u, v, w$ in terms of the basis vectors in the definition of $h_{u,v,w}$

$$h_{u,v,w}(c) = \mathbb{E}_{a,b}\left[\left(\sum \alpha_i \rho_i(a) + \sum \beta_i \rho_i(b)\right)^2 \left(\sum \gamma_i \rho_i(a \circ b \circ c)\right)\right]$$

and we expand this summation, all terms involving the trivial representation vector $\rho_1$ will equal zero since it is constant on all group elements. Furthermore, for terms of the form

$$\mathbb{E}_{a,b}[\rho_i(a)^2 \rho_k(a \circ b \circ c)] = \mathbb{E}_a[\rho_i(a)^2 \mathbb{E}_b[\rho_k(a \circ b \circ c)]] = 0$$

due to $\mathbb{E}_b[\rho_k(a \circ b \circ c)] = 0$ by orthogonality to the trivial representation vector.

Thus as was the case for the cyclic group, we study the term $\tilde{h}_{u,v,w}(c) := \mathbb{E}_{a,b}[u(a)v(b)w(a \circ b \circ c)]$ and derive an expression for the weighted margin in the following lemma.

**Lemma 16.** *Suppose the weights $\tau_c$ in the expression for the weighted margin in 10 were constant over conjugacy classes, i.e. we have $\tau_c = \tau_{C_i}$ for all $c \in C_i$ and $i \in [K]$. Then the weighted margin can be simplified as*

$$\sum_{m=2}^K \left(1 - \sum_{n=2}^K \frac{\tau_{C_n} |C_n| \chi_{R_m}(C_n)}{d_{R_m}}\right) \frac{\text{tr}(\boldsymbol{\alpha}_{R_m} \boldsymbol{\beta}_{R_m} \boldsymbol{\gamma}_{R_m}^T)}{d_{R_m}^2}.$$

*Proof.* Consider one term $\mathbb{E}_{a,b}[\alpha_i \beta_j \gamma_k \rho_i(a) \rho_j(b) \rho_k(a \circ b \circ c)]$ in the expansion of the product in $\tilde{h}_{u,v,w}(c)$. Note that $\rho_k(a \circ b \circ c)$ is one *entry* in the matrix of some irreducible representation evaluated at $a \circ b \circ c$; this can be expanded in terms of the same irreducible representation *matrix* evaluated at $a$, $b$, and $c$ using matrix multiplication. This results in terms of the form

$$\rho_i(a) \rho_j(b) \rho_{i'}(a) \rho_{j'}(b) \rho_{k'}(c)$$

in the expectation, where $\rho_{i'}$ and $\rho_{j'}$ correspond to entries of matrices from the same representation as $\rho_{k'}$. Thus if either $\rho_i$ or $\rho_j$ correspond to vectors from a *different* representation than $\rho_k$, the expectation of this term will be zero, by orthogonality of the basis vectors.

Hence we can assume that $\rho_i, \rho_j, \rho_k$ correspond to entries from the same representation $(V, R)$. Let $d = d_V$. Let us write $i = (i_1, i_2), j = (j_1, j_2), k = (k_1, k_2)$, the matrix indices for this representation. We can expand the term $\rho_k(a \circ b \circ c)$ as described above.

$$\rho_i(a)\rho_j(b)\rho_k(a \circ b \circ c) = \rho_i(a)\rho_j(b) \sum_{m=1}^{d} \rho_{(k_1,m)}(a \circ b)\rho_{(m,k_2)}(c)$$

$$= \sum_{\ell=1}^{d}\sum_{m=1}^{d} \rho_{(i_1,i_2)}(a)\rho_{(k_1,\ell)}(a)\rho_{(j_1,j_2)}(b)\rho_{(\ell,m)}(b)\rho_{(m,k_2)}(c).$$

From this it is clear that when taking the expectation over choosing $a, b$ uniformly, the only non-zero terms are when $(i_1, i_2) = (k_1, \ell)$ and $(j_1, j_2) = (\ell, m)$, once again by orthogonality of the basis vectors. Thus we have

$$\alpha_i\beta_j\gamma_k\rho_i(a)\rho_j(b)\rho_k(a \circ b \circ c) = \alpha_{(i_1,j_1)}\beta_{(j_1,j_2)}\gamma_{(i_1,k_2)}\rho^2_{(i_1,j_1)}(a)\rho^2_{(j_1,j_2)}(b)\rho_{(j_2,k_2)}(c).$$

Moreover, we know $\mathbb{E}[\rho_i(a)^2] = 1/d$, where $d$ is the dimensionality of the representation. Now, for a particular $c$, we will evaluate group all terms containing $\rho_{(j_2,k_2)}(c)$ and take the expectation over $a, b$, which yields

$$\frac{1}{d^2}\sum_{i_1=1}^{d}\sum_{j_1=1}^{d} \alpha_{(i_1,j_1)}\beta_{(j_1,j_2)}\gamma_{(i_1,k_2)}\rho_{(j_2,k_2)}(c). \tag{11}$$

From the third property of Lemma 15, for every conjugacy class $C_n$ for $n \in [K]$, we have

$$\sum_{c \in C_n} \frac{1}{d^2}\sum_{i_1=1}^{d}\sum_{j_1=1}^{d} \alpha_{(i_1,j_1)}\beta_{(j_1,j_2)}\gamma_{(i_1,k_2)}\rho_{(j_2,k_2)}(c) = 0$$

for $j_2 \neq k_2$. Thus, we can focus on diagonal entries $\rho_{(k,k)}(c)$ (i.e. where $j_2 = k_2$ in the expression 11 above). In this case, following directly from 11 grouping all terms containing $\rho_{(k,k)}(c)$ we get

$$\frac{1}{d^2}\sum_{i_1=1}^{d}\sum_{j_1=1}^{d} \alpha_{(i_1,j_1)}\beta_{(j_1,k)}\gamma_{(i_1,k)}\rho_{(k,k)}(c). \tag{12}$$

Note that this coefficient in front of $\rho_{(k,k)}(c)$ is the sum of the entries of the $k^{th}$ column of the matrix $(\boldsymbol{\alpha}_R\boldsymbol{\beta}_R) \odot \boldsymbol{\gamma}_R$ divided by $d^2$ (with $\boldsymbol{\alpha}_R\boldsymbol{\beta}_R$ interpreted as matrix product and $\odot$ being the Hadamard product). Recall that $i_1, j_1$, and $k$ are indices from the *same* representation $R$. By summing over all diagonal entries $(k, k)$ in $R$, we evaluate the expression

$$\sum_{i_1,j_1,k \in [d]} \frac{\alpha_{(i_1,j_1)}\beta_{(j_1,k)}\gamma_{(i_1,k)}}{d^2}\left[\rho_{k,k}(e) - \sum_{c \neq e}\tau_c\rho_{(k,k)}(c)\right]$$

$$= \frac{\text{tr}(\boldsymbol{\alpha}_R\boldsymbol{\beta}_R\boldsymbol{\gamma}_R^T)}{d^2}\left[1 - \sum_{n=2}^{K}\tau_{C_n}\sum_{c \in C_n}\rho_{(k,k)}(c)\right]$$

where we have replaced $\tau_c$ with the same weight $\tau_{C_n}$ for each non-trivial conjugacy class $C_2, \ldots, C_K$ and the term $\sum_{c \in C_n}\rho_{(k,k)}(c)$ is independent of the choice of $k$ (by property 4 of Lemma 15). Thus after summing over all $k \in [d]$ the coefficient in equation 12 is the sum of all entries of the matrix $(\boldsymbol{\alpha}_R\boldsymbol{\beta}_R) \odot \boldsymbol{\gamma}_R$ (which equals $\text{tr}(\boldsymbol{\alpha}_R\boldsymbol{\beta}_R\boldsymbol{\gamma}_R^T)$). Furthermore,

$$|C_n|\chi_R(C_n) = \sum_{c \in C_n}\sum_{k \in [d]}\rho_{(k,k)}(c) = \sum_{k \in [d]}\sum_{c \in C_n}\rho_{(k,k)}(c) = d\sum_{c \in C_n}\rho_{(k,k)}(c)$$

where the first equality follows from the definition of the character of the representation $R$ which is constant over elements in the same conjugacy class, and the last equality follows again from property 4 of Lemma 15. Thus $\sum_{c \in C_n}\rho_{(k,k)}(c) = |C_n|\chi_R(C_n)/d$ for all $k$.

Now we can evaluate our result for the weighted margin. The expression in 13 is the contribution of one representation $R$ to the total weighted margin. Thus by summing over all non-trivial representations of $G$, we get the final result.

$$\tilde{h}_{u,v,w}(e) - \sum_{c \neq e} w_c \tilde{h}_{u,v,w}(c) = \tilde{h}_{u,v,w}(e) - \sum_{n=2}^{K} \tau_{C_n} \sum_{c \in C_n} \tilde{h}_{u,v,w}(c) \tag{13}$$

$$= \sum_{m=2}^{K} \sum_{i_1,j_1,k \in [d_{R_m}]} \frac{\alpha_{(i_1,j_1)} \beta_{(j_1,k)} \gamma_{(i_1,k)}}{d^2} \left[ \rho_{k,k}(e) - \sum_{c \neq e} \tau_c \rho_{(k,k)}(c) \right] \tag{14}$$

$$= \sum_{m=2}^{K} \frac{\text{tr}(\boldsymbol{\alpha}_{R_m} \boldsymbol{\beta}_{R_m} \boldsymbol{\gamma}_{R_m}{}^T)}{d^2} \left[ 1 - \sum_{n=2}^{K} \tau_{C_n} \sum_{c \in C_n} \rho_{(k,k)}(c) \right] \tag{15}$$

$$= \sum_{m=2}^{K} \left[ 1 - \sum_{n=2}^{K} \frac{\tau_{C_n} |C_n| \chi_{R_m}(C_n)}{d_{R_m}} \right] \frac{\text{tr}(\boldsymbol{\alpha}_{R_m} \boldsymbol{\beta}_{R_m} \boldsymbol{\gamma}_{R_m}{}^T)}{d_{R_m}{}^2}. \tag{16}$$

$\square$

We have simplified the weighted margin expression for any set of weights on the conjugacy classes. Recall that we wish to optimize this weighted margin across individual neurons and then scale them appropriately to define the network $\theta^*$ satisfying **C.1** and Equation 1 to find the max margin solution.

The next lemma establishes the original $L_2$ norm restraint over neurons on the weighted margin problem in terms of the coefficients with respect to each representation.

**Lemma 17.** *The $L_2$ norm of $u$, $v$ and $w$ are related to the Frobenius norm of $\boldsymbol{\alpha}$, $\boldsymbol{\beta}$ and $\boldsymbol{\gamma}$ as follows:*

$$\|u\|^2 + \|v\|^2 + \|w\|^2 = \sum_{m=1}^{K} \frac{|G|}{d_{R_m}} \left( \|\boldsymbol{\alpha}_{R_m}\|_F^2 + \|\boldsymbol{\beta}_{R_m}\|_F^2 + \|\boldsymbol{\gamma}_{R_m}\|_F^2 \right)$$

*Proof.* The proof follows from 1st and 2nd point of Lemma 15. $\square$

By the above two lemmas, we want to maximize the weighted margin with respect to the norm constraint

$$\sum_{m=1}^{K} \frac{|G|}{d_{R_m}} \left( \|\boldsymbol{\alpha}_{R_m}\|_F^2 + \|\boldsymbol{\beta}_{R_m}\|_F^2 + \|\boldsymbol{\gamma}_{R_m}\|_F^2 \right) \leq 1. \tag{17}$$

Under this constraint, the following lemma provides the maximum value for the weighted margin, which occurs only when the weights $u, v, w$ are spanned by a single representation $R$.

**Lemma 18.** *Consider the set of representations $\mathcal{R}$ given by*

$$\mathcal{R} := \underset{m=2,..,K}{\arg\max} \frac{1}{\sqrt{d_{R_m}}} \left[ 1 - \sum_{n=2}^{K} \frac{\tau_{C_n} |C_n| \chi_{R_m}(C_n)}{d_{R_m}} \right]$$

*The weighted margin in Lemma 16 is maximized under the norm constraint in (17) only when the weights $u, v, w$ are spanned by a single representation belonging to the set $\mathcal{R}$; that is, $\boldsymbol{\alpha}_R, \boldsymbol{\beta}_R, \boldsymbol{\gamma}_R \not\equiv 0$ for only one non-trivial representation $R \in \mathcal{R}$, and are 0 otherwise. In this case, the maximum value attained is*

$$\frac{1}{3\sqrt{3}|G|^{3/2}} \max_{m=2,...,K} \frac{1}{\sqrt{d_{R_m}}} \left[ 1 - \sum_{n=2}^{K} \frac{\tau_{C_n} |C_n| \chi_{R_m}(C_n)}{d_{R_m}} \right].$$

*Proof.* First we consider the case where $u, v, w$ are spanned by only one representation. Then it suffices to evaluate

$$\max_{\boldsymbol{\alpha}_R, \boldsymbol{\beta}_R, \boldsymbol{\gamma}_R} \frac{\text{tr}(\boldsymbol{\alpha}_R \boldsymbol{\beta}_R \boldsymbol{\gamma}_R{}^T)}{d^2} \text{ s.t. } \left( \|\boldsymbol{\alpha}_R\|_F^2 + \|\boldsymbol{\beta}_R\|_F^2 + \|\boldsymbol{\gamma}_R\|_F^2 \right) \leq \frac{d}{|G|}.$$

Here let's denote the columns of $\boldsymbol{\alpha}_R$ (resp. $\boldsymbol{\beta}_R$) as $\vec{\alpha_j} = (\alpha_{j,1}, \ldots, \alpha_{j,d})$ for $1 \leq j \leq d$ (resp. $\vec{\beta_j}$). Thus $\text{tr}(\boldsymbol{\alpha}_R\boldsymbol{\beta}_R\boldsymbol{\gamma}_R^T) = \sum_{j,k}(\vec{\alpha_j} \cdot \vec{\beta_k})\gamma_{(j,k)}$. This can be viewed as the dot product of the linearizations of $\boldsymbol{\alpha}_R\boldsymbol{\beta}_R$ and $\boldsymbol{\gamma}_R$, and thus by Cauchy-Schwarz it follows that

$$\sum_{j,k}(\vec{\alpha_j} \cdot \vec{\beta_k})\gamma_{(j,k)} \leq \sqrt{\sum_{j,k}(\vec{\alpha_j} \cdot \vec{\beta_k})^2}\sqrt{\|\boldsymbol{\gamma}_R\|_F^2}$$

with equality when $\gamma_{(j,k)}$ is proportional to $\vec{\alpha_j} \cdot \vec{\beta_k}$. We can apply Cauchy-Schwarz once again to the first term on the right hand side above and obtain

$$\sqrt{\sum_{j,k}(\vec{\alpha_j} \cdot \vec{\beta_k})^2} \leq \sqrt{\sum_{j,k}\|\vec{\alpha_j}\|_2^2\|\vec{\beta_k}\|_2^2} = \sqrt{\|\boldsymbol{\alpha}_R\|_F^2\|\boldsymbol{\beta}_R\|_F^2}$$

once again with equality when all $\vec{\alpha_j}, \vec{\beta_k}$ are proportional to each other. Combining these together, we want to maximize $\sqrt{\|\boldsymbol{\alpha}_R\|_F^2\|\boldsymbol{\beta}_R\|_F^2\|\boldsymbol{\gamma}_R\|_F^2}$ subject to $\left(\|\boldsymbol{\alpha}_R\|_F^2 + \|\boldsymbol{\beta}_R\|_F^2 + \|\boldsymbol{\gamma}_R\|_F^2\right) \leq \frac{d}{|G|}$. By the AM-GM inequality, we have

$$\sqrt{\|\boldsymbol{\alpha}_R\|_F^2\|\boldsymbol{\beta}_R\|_F^2\|\boldsymbol{\gamma}_R\|_F^2} \leq \left(\frac{\|\boldsymbol{\alpha}_R\|_F^2 + \|\boldsymbol{\beta}_R\|_F^2 + \|\boldsymbol{\gamma}_R\|_F^2}{3}\right)^{3/2}$$

with equality when $\|\boldsymbol{\alpha}_R\|_F^2 = \|\boldsymbol{\beta}_R\|_F^2 = \|\boldsymbol{\gamma}_R\|_F^2 = \frac{d}{3|G|}$. Thus the maximum value attained is $\frac{1}{(|G|^{3/2}3\sqrt{3d})}\left[1 - \sum_{n=2}^K \frac{\tau_{C_n}|C_n|\chi_R(C_n)}{d_R}\right]$.

Now consider the general case when $u, v, w$ were spanned by the representations $R_2, \ldots, R_K$ (as $R_1$ does not appear in Equation 16). The norm constraint now becomes

$$\sum_{m=2}^K \frac{|G|}{d_{R_m}}\left(\|\boldsymbol{\alpha}_{R_m}\|_F^2 + \|\boldsymbol{\beta}_{R_m}\|_F^2 + \|\boldsymbol{\gamma}_{R_m}\|_F^2\right) \leq 1.$$

This can be equivalently written as

$$\|\boldsymbol{\alpha}_{R_m}\|_F^2 + \|\boldsymbol{\beta}_{R_m}\|_F^2 + \|\boldsymbol{\gamma}_{R_m}\|_F^2 \leq \frac{d_{R_m}\epsilon_m}{|G|} \quad \forall m \in \{2, \ldots, K\}$$

$$\epsilon_m \geq 0 \quad \forall m \in \{2, \ldots, K\}$$

$$\sum_{m=2}^K \epsilon_m \leq 1$$

Repeating the calculation above, we get that for a given $\epsilon_2, \ldots, \epsilon_K$, the maximum margin is given by

$$\sum_{m=2}^K \frac{\epsilon_m^{3/2}}{3\sqrt{3}|G|^{3/2}}\frac{1}{\sqrt{d_{R_m}}}\left[1 - \sum_{n=2}^K \frac{\tau_{C_n}|C_n|\chi_{R_m}(C_n)}{d_{R_m}}\right]$$

We want to maximize the expression above under the constraint that $\epsilon_m \geq 0 \quad \forall m \in \{2, \ldots, K\}$ and $\sum \epsilon_m \leq 1$.

Clearly, this is maximized only when one of the $\epsilon_i = 1$ and everything else is 0, with $i \in \arg\max_{m=2,\ldots,K} \frac{1}{\sqrt{d_{R_m}}}\left[1 - \sum_{n=2}^K \frac{\tau_{C_n}|C_n|\chi_{R_m}(C_n)}{d_{R_m}}\right]$. $\qquad\square$

Up until this point, we have kept our weighted margin problem generic without setting the $\tau_{C_n}$. If we naively chose $\tau_{C_n}$ to weigh the conjugacy classes uniformly, then the maximizers for this specific weighted margin would be only neuron weights spanned by the sign representation (of dimension 1). However, we cannot hope to correctly classify all pairs $a, b \in G$ using only the sign representation for our network $\theta^*$ and thus the maximizers for this weighted margin cannot be the maximizers for

the original max margin problem. The next lemma establishes an appropriate assignment for each $\tau_{C_n}$ such that the expression in Lemma 18 is equal for all non-trivial representations $R$, provided some conditions pertaining to the group are satisfied. Since the function $g \mapsto \tau_C$ (where $C$ is the conjugacy class containing $g$) is a class function, each $\tau_{C_n}$ can be expressed as a linear combination of characters $\chi_R(C_n)$.

**Lemma 19.** *For the group $G$ if we have $\sum_R d_R^{1.5} \chi_R(C) < 0$ for every non-trivial conjugacy class $C$, then the weights $\tau_{C_n}$ can be set as*

$$\tau_{C_n} = \sum_R z_R \chi_R(C_n)$$

*where $z_{R_{triv}} = 0$ and $z_R = \frac{d_R^{1.5}}{\sum_{m=2}^K d_{R_m}^{2.5}}$ otherwise, such that the maximum value from Lemma 18 is equal for all non-trivial representations $R$.*

*Proof.* Define the vectors $\tau$ and $\mathrm{char}(R)$ as

$$\mathrm{char}(R) = [\chi_R(C_1), \underbrace{\chi_R(C_2), \dots, \chi_R(C_2)}_{|C_2| \text{ times}}, \dots \underbrace{\chi_R(C_K), \dots, \chi_R(C_K)}_{|C_K| \text{ times}}],$$

$$\tau = [1, \underbrace{-\tau_{C_2}, \dots, -\tau_{C_2}}_{|C_2| \text{ times}}, \dots \underbrace{-\tau_{C_K}, \dots, -\tau_{C_K}}_{|C_K| \text{ times}}].$$

Then we can rewrite the max value of the weighted margin in Lemma 18 as

$$\frac{1}{|G|^{3/2} 3\sqrt{3d_R}} \left[ \frac{1}{d_R} \mathrm{char}(R)^T \tau \right] \tag{18}$$

for each non-trivial representation $R$. Since $\tau$ is a class function (viewed as a function on $G$), we can express $\tau$ as a linear combination $\tau = \sum_{n=1}^K z_{R_n} \mathrm{char}(R_n)$ of character vectors for each representation. By orthogonality, the inner product $\mathrm{char}(R)^T \tau = z_R$. Thus for the expression (18) to be equal for every non-trivial representation $R$, we require

$$z_R = d_R^{3/2} z_{R_{\mathrm{sign}}}.$$

Furthermore, since $1 - \sum_{n=2}^K \tau_{C_n} = 0$ and $\mathrm{char}(R_{\mathrm{triv}})$ is a vector with strictly positive values that is orthogonal to all other character vectors, we must have $z_{R_{\mathrm{triv}}} = 0$. To solve for $z_{R_{\mathrm{sign}}}$, since the first component of $\tau$ equals 1 and $\chi_R(C_1) = d_R$ for all $R$, we have

$$\sum_{m=2}^K z_{R_m} d_{R_m} = \sum_{m=2}^K d_{R_m}^{2.5} z_{R_{\mathrm{sign}}} = 1 \implies z_{R_{\mathrm{sign}}} = \sum_{m=2}^K d_{R_m}^{2.5}.$$

To conclude the proof, note that we need the weights $\tau_{C_n}$ to be positive; this is guaranteed as long as for each conjugacy class $C$, we have $\sum_{n=2}^K \chi_{R_n}(C) d_{R_n}^{3/2} < 0$ (recall the entries of $\tau$ being $-\tau_{C_n}$).

$\square$

Up until now, we have established a weighted margin problem and proven that the neurons which maximize this are spanned by only one representation out of any of the non-trivial representations. Now we give a precise construction of the neuron weights $u, v, w$ such that they implement $\mathrm{tr}(R(a)R(b)R(c)^{-1})$ for all inputs $a, b \in G$ and outputs $c \in G$ for a given representation $R$. These neuron weights expressed in terms of the basis vectors will have coefficients that also maximize $\mathrm{tr}(\boldsymbol{\alpha}_R \boldsymbol{\beta}_R \boldsymbol{\gamma}_R^T)$.

**Lemma 20.** *For every non-trivial representation $R$, there exists a construction of the network weights such that given inputs $a, b \in G$, the output at $c$ is $\mathrm{tr}(R(a)R(b)R(c)^{-1})$ using $2d_R^3$ neurons and the corresponding coefficients $\boldsymbol{\alpha}_R, \boldsymbol{\beta}_R, \boldsymbol{\gamma}_R$ for each neuron achieve the maximum value $\mathrm{tr}(\boldsymbol{\alpha}_R \boldsymbol{\beta}_R \boldsymbol{\gamma}_R^T) = (d_R/3|G|)^{3/2}$.*

*Proof.* Since the representations are unitary, we have

$$\text{tr}(R(a)R(b)R(c)^{-1}) = \text{tr}(R(a)R(b)R(c)^T) = \sum_{i,j,k} R(a)_{(i,j)}R(b)_{(j,k)}R(c)_{(i,k)}$$

and thus it suffices to show how to obtain $R(a)_{(i,j)}R(b)_{(j,k)}R(c)_{(i,k)}$ with a combination of neurons. For this, set one neuron's coefficients to equal $\alpha_{(i,j)} = \beta_{(j,k)} = \gamma_{(i,j)} = 1/\sqrt{3|G|}$ and 0 otherwise. Then the output given $(a, b)$ at $c$ is

$$\frac{(R(a)_{(i,j)} + R(b)_{(j,k)})^2 R(c)_{(i,k)}}{(3|G|)^{3/2}}.$$

Set another neuron's coefficients to equal $\alpha_{(i,j)} = 1/\sqrt{3|G|}, \beta_{(j,k)} = \gamma_{(i,k)} = -1/\sqrt{3|G|}$. Then the sum of the outputs of these two neurons at $c$ is precisely

$$\frac{R(a)_{(i,j)}R(b)_{(j,k)}R(c)_{(i,k)}}{(3|G|)^{3/2}}.$$

Thus we need $2d_R^3$ neurons to create the summand for each $i, j, k$ to implement $\text{tr}(R(a)R(b)R(c)^{-1})$. This construction also satisfies $\text{tr}(\boldsymbol{\alpha}_R\boldsymbol{\beta}_R\boldsymbol{\gamma}_R^T) = (d_R/3|G|)^{3/2}$ for every neuron. $\qquad\square$

Once we have defined these neuron constructions, it only remains to scale these optimal neurons appropriately as given in Lemma 8 such that we can construct our final network $\theta^*$ satisfying condition **C.1** and Equation 1.

**Lemma 21.** *Given the network given in Lemma 20, for every neuron spanned by non-trivial representation $R$ we scale the weights $u, v, w$ by $d_R^{1/3}/\Delta$, where $\Delta$ is a constant normalization term such that the norm constraints of the max margin problem still hold. Then the expected output of any element contained in any non-trivial conjugacy class $C$ for inputs $a, b$ is $-1/\Delta^3$, i.e. the output is equal for all conjugacy classes.*

*Proof.* For a given neuron spanned by a non-trivial representation $R$, we know that its output for at $c$ for each input pair $(a, b)$ is $\chi_R(abc^{-1}) = \chi_R(C)$ where $C$ is the conjugacy class containing $abc^{-1}$. After scaling each weight by $d_R^{1/3}/\Delta$, the corresponding output is scaled by $d_R/\Delta^3$. Due to column orthogonality of the characters with the trivial conjugacy class (i.e. $\sum_{n=1}^K \chi_{R_n}(e)\chi_{R_n}(C) = 0$ for for all non-trivial conjugacy classes $C$), this output simplifies to

$$\sum_{n=2}^K \frac{d_{R_n}\chi_R(C)}{\Delta^3} = -\frac{1}{\Delta^3}\sum_{n=2}^K \frac{d_{R_n}\chi_R(C)}{\Delta^3} = -\frac{1}{\Delta^3}, \tag{19}$$

which is constant for all non-trivial conjugacy classes $C$.

$\qquad\square$

With this lemma, we define the network $\theta^*$ according to this scaling and guarantee that it satisfies **C.1** and Equation 1. Applying Lemma 8 gives us our final result that the solutions for the max margin problem have the desired properties in Theorem 7.

### J.1 PROOF THAT ALL REPRESENTATIONS ARE USED

This proof follows exactly the same argument as for the modular addition case (Section G.3).

For this proof, we will introduce the multidimensional Fourier transform for groups. For a function $f : G^3 \to \mathbb{R}$, this is defined as

$$\hat{f}(j, k, l) = \sum_{a \in |G|} \rho_j(a) \sum_{b \in |G|} \rho_k(b) \sum_{c \in |G|} \rho_l(c)f(a, b, c)$$

Similar to the modular addition case, for a single margin maximizing neuron, we know it uses only one of the representations for input and output neurons, let's say $R_m$. Then, considering just the basis vectors with respect to $R_m$, we can say, that the output of this neuron is given by

$$f(a, b, c) = \left[ \sum_{i \in d_{R_m}} \sum_{j \in d_{R_m}} \alpha_{(i,j)} \rho_{(i,j)}[a] + \beta_{(i,j)} \rho_{(i,j)}[b] \right]^2 \left( \sum_{k \in d_{R_m}} \sum_{l \in d_{R_m}} \gamma_{(k,l)} \rho_{(k,l)}[c] \right)$$

Now, for the squared terms, these are either dependent on $a, c$ or $b, c$. These have non-zero fourier coefficients only if $j = 0$ or $k = 0$.

For the cross terms, by orthogonality of the representatons, we can say, if either $j, k$ or $l$ does not belong to $R_m$, then $\hat{f}(j, k, l) = 0$.

Thus, for a single neuron, $\hat{f}(j, k, l)$ is only non-zero if $j = 0$, $k = 0$ or if $j, k$ and $l$ belong to the same representation.

Independent of the above considerations, we know by Lemma 9 that the function $f$ implemented by the network has equal margin across different inputs and across different classes for the same input. In other words, $f$ can be decomposed as

$$f(a, b, c) = f_1(a, b, c) + f_2(a, b, c)$$

where

$$f_1(a, b, c) = F(a, b)$$

for some $F : G \times G \to \mathbb{R}$, and

$$f_2(a, b, c) = \lambda \cdot \mathbf{1}_{a \circ b = c}$$

where $\lambda > 0$ is the margin of $f$.

The Fourier transform of $f_1$ is

$$\hat{f}_1(j, k, l) = \begin{cases} \hat{F}(j, k) & \text{if } \ell = 0 \\ 0 & \text{otherwise} \end{cases}$$

For $f_2$, consider the expression of the fourier transform:

$$\hat{f}_2(j, k, l) = \lambda \sum_{a \in |G|} \rho_j(a) \sum_{b \in |G|} \rho_k(b) \rho_l(a \circ b)$$

Now, $\rho_l(a \circ b) = \sum \rho_{l'}(a) \rho_{k'}(b)$ for some $j', k'$ given by the relation that $R(a \circ b) = R(a)R(b)$, where $R(a)R(b)$ denoted the matrix product of $R(a)$ and $R(b)$. Now, clearly if $j, k$ and $l$ belong to different representations, then $\hat{f}_2(j, k, l)$ is 0. For $j, k, l$ belonging to the same representation, $\hat{f}_2(j, k, l)$ will be non-zero whenever $j = j'$ and $k = k'$ (or $j = k'$ and $k = j'$), and the value will be given by $\lambda |G|^2 / d_{R_m}^2$. Thus, $f = f_1 + f_2$ has support on all the representations.

But, this is only possible if there is atleast one neuron for each representation, as a single neuron places non-zero fourier mass only on one of the representation.

## J.2 A GENERAL THEOREM FOR FINITE GROUPS

As mentioned in section 6, Theorem 7 does not hold for all groups because of the required condition that $\sum_{n=2}^{K} d_{R_n}^{1.5} \chi_{R_n}(C) < 0$ for every non-trivial conjugacy class. Recall that in the previous section, we had to define an appropriate weighting over *all* conjugacy classes such that the margin of a neuron did not scale down with the dimension of the neuron's spanning representation. We also had to define an appropriate scaling over *all* representations so we could use the neuron maximizers of the weighted margin to construct a network $\theta^*$ to invoke Lemma 8. This is akin to selecting the entire character table for our margin analysis; in this section, we show how our analysis is amenable to selecting a *subset* of the character table for the margin analysis of a general finite group $G$, which can lead to a max margin solution in the same way as above. This will occur upon solving a system of two linear equations, as long as these solutions satisfy some conditions.

Namely, let $\kappa_R, \kappa_C \subset [K] \setminus \{1\}$ be subsets indicating which representations and which conjugacy classes will be considered in the scaling and weighting respectively, with $|\kappa_R| = |\kappa_C|$. If we view the character table as a matrix and consider the square submatrix pertaining to only the representations indexed by elements in $\kappa_R$ and the conjugacy classes indexed by elements in $\kappa_C$, the rows are $\chi_{R_m}$ for fixed $m \in \kappa_R$ and the columns are $[\chi_{R_m}(C_n)]_{m \in \kappa_R}$ for fixed $n \in \kappa_C$.

Instead of requiring expression (18) to be equal for all representations in the proof of Lemma 19, we can instead require that they are equal across representations in $\kappa_R$. To be precise, consider the following set of equations over variables $\tau_{C_n}, n \in \kappa_C$:

$$\left(1 - \sum_{n \in \kappa_C} \frac{\tau_{C_n}|C_n|\chi_{R_m}}{d_{R_m}}\right) = \sqrt{\frac{d_{R_m}}{d_{R_{m'}}}} \left(1 - \sum_{n \in \kappa_C} \frac{\tau_{C_n}|C_n|\chi_{R_{m'}}}{d_{R_{m'}}}\right) \ \forall \ m, m' \in \kappa_R,$$

$$\sum_{n \in \kappa_C} \tau_{C_n} = 1.$$

This gives a system of $|\kappa_C|$ linear equations in $|\kappa_C|$ variables. Let the solution be denoted as $\tau^*_{C_n}$ for each $n \in \kappa_C$.

Furthermore, just as we established in equation 19, we can identify a scaling dependent on each representation such that the output remains constant for all conjugacy classes in $\kappa_C$ and such that if we had used this scaling for neurons maximizing the weighted margin, the $L_{2,3}$ norm constraint is maintained. This can be represented using the following set of equations with variables $\lambda_{R_m}$:

$$\sum_{m \in \kappa_R} \lambda_{R_m} \chi_{R_m}(C_n) = \sum_{m \in \kappa_R} \lambda_{R_m} \chi_{R_m}(C_{n'}) \ \forall \ n, n' \in \kappa_C$$

$$\sum_{m \in \kappa_R} \lambda_{R_m} = 1.$$

This again forms a system of $|\kappa_R|$ linear equations in $|\kappa_R|$ variables. Let the solution be denoted as $\lambda^*_{R_m}$. Suppose the following conditions are satisfied:

1. The weighting and scaling are positive: $\lambda^*_{R_m}, \tau^*_{C_n} \geq 0$ for all $m \in \kappa_R, n \in \kappa_C$.

2. For any $m \in \kappa_R$ and $m' \notin \kappa_R$, we have

$$\left(1 - \sum_{n \in \kappa_C} \frac{\tau_{C_n}|C_n|\chi_{R_m}}{d_{R_m}}\right) \geq \sqrt{\frac{d_{R_m}}{d_{R_{m'}}}} \left(1 - \sum_{n \in \kappa_C} \frac{\tau_{C_n}|C_n|\chi_{R_{m'}}}{d_{R_{m'}}}\right).$$

3. For any $n \in \kappa_C$ and $n' \notin \kappa_C$, we have

$$\sum_{m \in \kappa_R} \lambda_{R_m} \chi_{R_m}(C_n) \geq \sum_{m \in \kappa_R} \lambda_{R_m} \chi_{R_m}(C_{n'}).$$

The second condition ensures that the representations in $\kappa_R$ indeed maximize the weighted margin, and no other representations maximize it. The third condition above ensures that the conjugacy classes in $\kappa_C$ are on the margin, and no other conjugacy class can be on the margin. Then it follows that neurons spanned by the representations in $\kappa_R$ will maximize the weighted margin defined using $\tau^*$ with all conjugacy classes in $\kappa_C$ on the margin, and thus scaling these neurons by $\lambda^*$, we have a network $\theta^*$ that is a max margin solution for the group $G$.

