# OpenReview forum: "Feature emergence via margin maximization: case studies in algebraic tasks"
_ICLR.cc/2024/Conference — ICLR 2024 spotlight_

### Official Review · Reviewer_hFRW · 2023-10-23

**Soundness:** 4 excellent
**Presentation:** 2 fair
**Contribution:** 4 excellent
**Rating:** 6
**Confidence:** 3

**Summary:**

The authors propose a mathematical framework to explain why some specific algorithms are implemented inside neural networks that were trained on some algebraic tasks. Firstly, they use the result from Wei et al. (2019a) to argue that minimizers of the $L_r$ regularized cross-entropy loss have a maximal margin. Using the proof blueprint introduced in Section 3, they can characterize precisely the internal representations learned when a specific architecture is trained on a specific task. They consider three concrete algebraic tasks: Modular addition, sparse parity and finite groups with unitary, irreducible and real representations. The theoretical findings for these three examples are confirmed in controlled experiments.

Summary of my review: I think the results presented in this work are very interesting and form an important step toward understanding why specific algorithms emerge in neural networks. I found the paper challenging to read, especially Section 3. Part of it might be attributable to the fact that I’m not an expert in this field, but I nevertheless believe that the writing could be improved (see the concrete suggestion I made below). For this reason, I give a score of 6.

**Strengths:**

- The framing of the paper is good. I very much like the direction of explaining mathematically why some specific “algorithms” emerge when some architectures are trained on some dataset. I think this is important.
- I found the theoretical results very interesting and surprising. They form an important step towards the stated goal.
- The empirical validation for each special case was convincing.
- I am not well versed in this area so it is very difficult for me to assess novelty.

**Weaknesses:**

- I noticed a few times that notations/terms were introduced without proper definitions. Maybe these are well-known in the field, but I think they should be restated for outsiders like me. I also found that some things should be deduced from context and would gain in being stated more explicitly. A few examples:
    - In Section 2, it would be helpful to state that the output of the neural network is d-dimensional somewhere. Same for the neuron. I am used to having neurons outputting a scalar, but here $\phi$ is $\mathbb{R}^d$-valued, right?
    - Section 3: $\Delta(\mathcal{Y})$ designate the set of functions $f: \mathcal{Y} \rightarrow [0,1]$ such that $\sum_{y \in \mathcal{Y}}f(y)$, right? Might be worth making it more explicit.
    - Might be worth explicitly saying: $\text{spt}(q)$ denotes the support of $q$ in C.1.
    - Section 6.1: $R_n$ for $n=1…K$ are not defined.
- I appreciate that the authors made an effort to provide details about the proofs in the main text, especially in Section 3 which provides blueprints for all results. That being said, I found this section very difficult to digest (more specifically, everything that comes after Theorem 1).
    - Something that would help me is to state the quantifiers (like “for all” and “there exists”) more explicitly. For example, does the first implication of Lemma 2 hold for all $\theta^*$ and $q^*$?
    - How strong are conditions C1, C2, C3? I’m not sure I could summarize what they mean after rereading multiple times. I think my confusion is related to quantifiers again.
    - Lemma 3: Under “Single neuron optimization”, what is a “directional support”? Under “Using multiple neurons”, I’m confused about the quantifier for the $\lambda_i$’s. Is it “for all” or “there exists”?
- I think it would be better if the specifics of the architectures used for training in each special cases (Sections 4, 5, 6) were given in the relevant sections themselves instead of before the rather long and technical Section 3.
- Proof outline of Theorem 4: The part about the constrained optimization leading to sparsity in Fourier space went over my head. Where does this constraint come from? The norm constraint in the max-margin problem? I understand this is a proof sketch, but I think this could be made more transparent without giving all the details.

Minor:
- $\\{u,v,w\\} \in \mathbb{R}^d$ is unclear. Shouldn’t it be $u,v,w \in \mathbb{R}^d$?
- Proof outline of Theorem 4: I thought the notation $\mathbb{E}_{a,b}[\psi’(u,v,w)]$ to be a bit confusing and “too different” from the way it was introduced in Section 3.1.

**Questions:**

See above.

---

> ### Author Response · Authors · 2023-11-20
> **Response to Reviewer hFRW**
>
> We thank the reviewer for their comments and are happy to see that they find the direction of the paper important, and the theoretical results interesting and surprising. The specific comments are addressed below:
>
> 1. **Notations and Section 3** - See general comments regarding our revisions to section 3. We appreciate the specific comments and have updated section 2 to introduce notation more clearly.
> 2. **Proof outline of Theorem 4 (Theorem 5 in the revised version)** - Thanks for the suggestions. The norm constraint indeed comes from the max-margin problem, we now state this in the proof outline.
> 3. **Minor comments** - Indeed, each of $u, v, w \in \mathbb{R}^d$ and we have made this revision. We have also fixed the notation for the expected class-weighted margin in the proof outline of Theorem 4 (Theorem 5 in the revised version) to be consistent with section 3.
>
> We hope that we have adequately addressed all of your concerns regarding clarity of exposition in our work. We would be happy to answer any additional questions you have.

---

> > ### Comment · Reviewer_hFRW · 2023-11-22
> >
> > Thanks for your answers!

---

### Official Review · Reviewer_xfNV · 2023-10-27

**Soundness:** 3 good
**Presentation:** 3 good
**Contribution:** 3 good
**Rating:** 8
**Confidence:** 3

**Summary:**

The paper shows a 2-layer NN with homogeneous polynomial activations can learn modular addition, parity, and finite group operation by max-margin with SGD. For modular addition, the max-margin solution indeed utilizes Fourier features and cover all frequency. Thus, they show an implicit bias and argue that NN prefers to learn Fourier-based circuits, rather than other potential circuits capable of executing the same function.

**Strengths:**

The paper has two contributions.
- Generally speaking, it is hard to find all max-margin solutions for general data distributions and general networks. The paper introduces a general framework to analyze the homogeneous NN max margin solution under some distribution with symmetric properties. By Lemma 2 and Lemma 3, we can solve network-level optimal solutions by solving neuron-level search. Thus, we can simplify the optimization problem from a max-min problem over margin functions to a max problem over linear combinations of neurons. As far as I know, this method is novel and useful. It may be powerful to help the NN loss landscape. Also, see the related question in Questions.
- The paper studies three case studies by utilizing the above framework to show how powerful it is. In particular, for the module addition part, the observation about the max-margin solution being Fourier features and covering all frequencies is insightful.

The writing is good and the motivation is clear.

**Weaknesses:**

- The paper asked, “Why does the network **consistently** prefer such Fourier-based circuits, amidst other potential circuits capable of executing the same function?” in the introduction section. However, it seems when learning modular addition the NN prefers Fourier-based circuits, while learning parity or finite group operation may not (correct me if I am wrong). Thus, it seems to mislead the reader's expectations. Moreover, it would be good to discuss more here about other potential circuits. (1) For modular addition, what are else potential circuits? Some discussion and comparison would be good. (2) Is there any other data distribution that NN prefers to learn other potential circuits rather than Fourier-based circuits? Some more case studies or illustrations would be good.
- The paper can only handle homogeneous functions as it needs to use the conclusion from [1]. Then, the neural network cannot have the bias terms. This is restricted. For example, Theorem 5 needs $\Omega(2^k)$ number of neurons to learn sparse parity, while a 2-layer ReLU NN with bias only needs $\Omega(poly(k))$ neurons to learn structured sparse parity [2,3]. This weakness may be beyond the scope of this paper but it is important.
- As the conclusion section mentioned, the paper needs some properties about homogeneous so that it can only handle $L_{2, \nu}$ norm regularization. It would be good to extend to more general settings.

[1] Kaifeng Lyu and Jian Li. Gradient Descent Maximizes the Margin of Homogeneous Neural Networks. ICLR 2020.

[2] Amit Daniely and Eran Malach. Learning parities with neural networks. NeurIPS 2020.

[3] Zhenmei Shi, Junyi Wei, and Yingyu Liang. Provable Guarantees for Neural Networks via Gradient Feature Learning. NeurIPS 2023.

**Questions:**

From my perspective, Lemma 2 works for any homogeneous parametric function while Lemma 3 only works for homogeneous 2-layer NN. If my understanding is correct, can we generalize Lemma 3 to any homogeneous parametric function? If not, what is the problem preventing us?

---

> ### Author Response · Authors · 2023-11-20
> **Response to Reviewer xfNV**
>
> We thank the reviewer for their comments, and are happy to see that the reviewer finds the techniques novel and useful, and the overall work insightful. The specific comments are addressed below:
>
> 1. **Preference to Fourier-based circuits** - We will clarify that the quoted line “Why does the network consistently prefer such Fourier-based circuits”, is referring specifically to the modular addition task. We are not claiming that there is any bias towards Fourier features on other tasks. Rather, this phenomenon is an adaptation to the very specific structure of the modular addition task. Regarding other potential circuits: in Appendix D, we describe a circuit which does not use Fourier features but achieves perfect accuracy on the modular addition task.
> 2. **Homogeneous architectures** - Yes, our approach can only handle homogeneous architectures, but that still allows us to have biases in the first layer (which are still homogeneous in the weights). The requirement for 2^k neurons comes from the fact that we are using a quadratic activation function. We too hope that our theory can be extended in future work to handle more general regularizations and activation functions.
> 3. **Generalizing Lemma 3** - Yes, Lemma 3 can be generalized to any homogeneous parametric function having an additive structure, i.e, the output is given by the sum of individual homogeneous functions (which do not share parameters). However, a neural network having more than one hidden layer does not have this additive structure, as some of the hidden layers are shared.

---

> > ### Comment · Reviewer_xfNV · 2023-11-22
> >
> > Thanks for your response. It answers my question clearly and well. I have read all reviewers' comments and tend to keep my score.

---

### Official Review · Reviewer_jPZB · 2023-11-01

**Soundness:** 4 excellent
**Presentation:** 3 good
**Contribution:** 2 fair
**Rating:** 8
**Confidence:** 2

**Summary:**

In this paper, the authors present a theoretical development to analyze why a neural network with one hidden layer when trained on some algebraic tasks using SGD reliably and consistently decomposes the problem as a Fourier type of decomposition.

The authors focus on the max-margin classifiers and present a the conditions and a construction from single to many neurons for the fourier representation. The authors show the margin required for the decomposition and show how with a quadratic activation the neurons achieve single frequencies.

The authors show empirically what their proofs demonstrate.

**Strengths:**

The paper is technically sound and well written. Although it is very specialized and not easy to read for the general practitioner of deep learning.
The empirical evaluation confirms the behavior at maximum margin conditions and demonstrates as well what the theory predicted.

**Weaknesses:**

The paper specializes on a particular phenomena and on a restricted architecture. It is great that they found a connection on why this representation works, but it is not clear how this could extend to other architectures or use cases.

**Questions:**

I had two questions. The first one is regarding what you mention regarding the optimizer. Is this a behavior that would only be observed with SGD? I can imagine that coordinate descent would achieve the same results if one things about the construction from single neurons to whole network. But I can imagine other optimizers also behaving well and this behavior being more a characteristic of the loss function and the activation function?

The second one is, this paper reminds me of the positional encoding of language models. You are showing that the network can decompose groups via frequency based representations. I'm wondering if the same is not indirectly happening at the first layers of a language model, i.e., whether this behavior that you observe could happen also at the level of layers and not just over the wide network you constructed.

---

> ### Author Response · Authors · 2023-11-20
> **Response to Reviewer jPZB**
>
> We thank the reviewer for their comments. The specific comments are addressed below:
>
> 1. **Quadratic MLP** - See our general comment regarding the use of our simplified architecture.
> 2. **SGD** - Thanks for the point about SGD. Indeed the result will also work for other descent methods, as long as the methods reach the global optimum of the loss. We have clarified this in the introduction.
> 3. **Positional Encodings** -  The connection to positional encodings of language models is intriguing! We think that in general exploring how neural networks adapt to symmetries and algebraic structure in training data is a very interesting direction.

---

> > ### Comment · Reviewer_jPZB · 2023-11-22
> >
> > thanks for your answers

---

### Official Review · Reviewer_CYd2 · 2023-11-04

**Soundness:** 4 excellent
**Presentation:** 2 fair
**Contribution:** 2 fair
**Rating:** 6
**Confidence:** 3

**Summary:**

The paper studies the optimal representation for a feedforward neural network that allows for perfect generalization on arithmetic tasks such as modular addition, parity, and other finite group operations. It falls under the recently established field of mechanistic interpretability. The theoretical result shows that a well-trained model uses Fourier features to perform modular addition and the learned features correspond to an irreducible group representation of the input data. Previous works discover that the learned model leverages a particular algorithm based on Fourier features to perform modular arithmetic; this work offers a partial explanation as to why such a strategy is learned, in particular why Fourier features are preferred.

The first theoretical result defines a class-weighted margin and studies a one-hidden-layer neural network with squared activation trained with cross-entropy loss and norm regularization.  First,  the paper states 3 conditions for the data and margin. Under these conditions, the paper establishes a reduction from the maximization of the margin to the maximization of the class-weighted margin (lemma 2). Then, the paper establishes a reduction for full parameter search to the search of individual neuron $\omega_i$.

The second theoretical result demonstrates that under these assumptions, the max-margin solution for modular addition on the cyclic group has an analytical max margin and its features are the Fourier features (Theorem 4). Similarly,  the max-margin solution for sparse parity also has an analytical max margin and different features. Finally, the paper presents a general recipe for tha max-margins solution for group composition (Theorem 6).
he c
Experiments show that the margins of the model indeed reach the value predicted by the theory, thus validating the theoretical results.

**Strengths:**

The paper attempts to tackle an important problem in mechanistic interpretability on synthetic arithmetic tasks. The original findings of Nanda et al were very surprising. This paper is able to shed some light on the fact that the seemingly bizarre algorithm in Nanda et al. is not so strange after all. More importantly, the paper provides a general recipe beyond those studies in Nanda et al., which I find to be technically interesting. The empirical results match the prediction of the theorem extremely well (almost exactly) so even though I have not checked the proof thoroughly, I am reasonably confident that the technical results are correct. Overall, I think this paper is a nice addition to the emergent field of mechanistic interpretability.

**Weaknesses:**

While this paper presents some interesting results, I feel that there is still a somewhat big gap between the theoretical results and the reality. I believe in general it is not clear how studying simple task like this will lead us to deeper understanding

- The setting studied in the paper is an extremely simple MLP that is very different from the one studied by Nanda et al.
- The results and notations in the paper are somewhat difficult to follow (see some of my questions below). I feel the paper's clarity would benefit from having a more high-level summary of each of the theoretical results and assumptions and their intuitive implications. For example, I find the assumptions C.1-C.3 difficult to follow. It may also be a good idea to have a visual illustration of some of these results.
- If I understand correctly, it seems like the training is done on all of the data in the support so there is no generalization needed and all analysis is done on the representation, so it does not address one of the most shocking observations in these arithmetic tasks, namely, grokking.
- There is no optimization analysis so the whole picture is not complete (although we do see that the converged result matches the prediction).

**Questions:**

- What is the relationship between $u,v,w$ and $\omega$? Concatenation?
- What is $P(D)$ in Sec 3? Is it an arbitrary distribution?
- Where does $\tau(x,y)$ in Sec 3 come from?
- I'm not sure if I understand C1. How can the support of the distribution be the subset of argmin which is presumably a single point or a set of discrete points ($\text{sqt}(q^*)\subseteq \text{argmin}_{(x,y) \in D} g(\theta^*, x, y)$)? Does that mean $q^*$ is a discrete distribution on the $\mathbb{R}^n$?
- One impression I get from reading this paper is that the provided construction actually seems to be the only sensible solution given the constraints of the function class. Is this a correct assessment? This is an important statement because it is somewhat similar to saying that we already assume the answer by making all the assumptions, which further undermines these frameworks' ability to help us understand real LLMs. If not,  what other circuits are possible beyond the constructions in the paper? How do we know this is not the only way to do it?

---

> ### Author Response · Authors · 2023-11-20
> **Response to Reviewer CYd2**
>
> We thank the reviewer for their comments. Specific comments are addressed below:
>
> 1. **Quadratic MLP** - See our general comment regarding the use of our simplified architecture.
> 2. **Section 3 notations** - See our general comment regarding our revisions to section 3.
> 3. **Generalization** - Yes, we do agree that the main focus of the work is on representation learning, and not generalization. We do not claim to be providing an explanation for grokking, which is an independently interesting phenomenon.
> 4. **Optimization** - Yes, we agree that we have only provided the maximum margin result, which when combined with the result of Wei et al, shows that the global optimum of the training loss is indeed the maximum margin solution for vanishing regularization. We do not prove that the global optimum will be reached by training; as you note, we provide empirical evidence instead.
> 5. **Relation between $\omega$ and $u,v,w$** - Yes $\omega$ is the concatenation of $u,v,w$.
> 6. **Definition of P(D)** - P(D) is the set of distributions on the samples belonging to the training dataset.
> 7. **Intuition about $\tau(x,y)$** - Please look at the updated section 3, where we provide an intuition about $\tau(x,y)$. This is required to convert the network-level search problem to a neuron-level search for multi-class classification.
> 8. **Support of q** - Yes, $q^*$ is discrete as it can only have support on the points in the training dataset $D$.
> 9. **Other solutions** - Regarding whether the provided construction is the only sensible solution—please look at Appendix D in the paper. We have provided another construction which does not display any Fourier sparsity; so there are other solutions possible, but margin maximization favors the Fourier-sparse solutions. We have added a reference to it after the informal theorem.
>
> We emphasize that our work is the first to provide a rigorous explanation for Fourier feature emergence (and representation theoretic feature emergence) in any setting. Although we do not provide a complete explanation of optimization on these tasks for general architectures, we believe that our work is a substantial step forward towards a more comprehensive understanding of the various surprising phenomena that occur for algebraic tasks.

---

> ### Comment · Reviewer_CYd2 · 2023-11-22
>
> I thank the authors for their reply and revision of the paper. Most of my questions have been addressed. Regarding the architecture, I don't fully agree that simple MLP is enough for this case. While the problem studied in this work is very important, the most important aspect of Nanda et al, in my opinion, is about grokking/generalization. For this purpose, it is not clear whether attention is needed for the phenomena to happen or if the same phenomena can occur for MLP. I think either case would be interesting.
>
> Currently, I think the paper could mislead people to think that the result pertains to grokking/generalization since it studies max margins solution which traditionally helps with generalization. I would encourage the authors to make this more explicit. In any case, I think this is a very neat result and it should be accepted.

---

> > ### Author Response · Authors · 2023-11-22
> >
> > Thanks for the response. In our introduction, we do already emphasize that while max margin analysis is typically used to understand generalization, we are using it for a different purpose. In order to make this even clearer, we have added the word "instead" to the following passage:
> >
> > *Most work on inductive bias in deep learning is motivated by the question of understanding why networks generalize from their training data to unobserved test data. It can be non-obvious how to apply the results from this literature to **instead** understand what solution will be found when a particular architecture is trained on a particular type of dataset.*
> >
> > Regarding whether MLPs can exhibit grokking, there are multiple previous works ([1],[2]) demonstrating that the phenomenon is indeed replicated for MLPs (even with quadratic activations). So, studying transformers is not essential for studying grokking.
> >
> > [1] - Andrey Gromov, Grokking Modular Arithmetic, https://arxiv.org/abs/2301.02679
> >
> > [2] - Ziming Liu et al. - Towards Understanding Grokking: An Effective Theory of Representation Learning, NeurIPS 2022

---

### Author Response · Authors · 2023-11-20
**General comments to reviewers**

We thank all the reviewers for providing insightful comments and feedback for our work. We address common concerns and summarize the revisions we have made to the submission below.

1. **Regarding the use of our simplified architecture:** We do study a 1-hidden layer MLP with quadratic activations, which is different from the architecture (1-layer transformer) studied in Nanda et al. However, other works (such as Zhong et al. and Gromov) have shown that Fourier features still emerge in one-hidden layer MLP with ReLU/quadratic activation. Arguably, using a transformer is “overkill” for this task, because the sequences are only of length 2. Given that the phenomenon holds more generally than just 1-layer transformers, we do not think it is necessary to study it only on this architecture, and indeed, proofs in simpler settings can provide more intuitive understanding. That being said, we think the question of how our results can be extended to other architectures is very interesting.

2. **Regarding section 3:** We agree that section 3 was difficult to follow. In the revised version of the paper, we have completely rewritten this section to be more reader-friendly and moved more cumbersome notation to the appendix. Specifically, we outline our general approach using the simpler case of binary classification, and mention how this can be extended to multi-class classification (which is necessary for modular addition and finite group composition) by introducing the notion of class-weighted margin. We hope that this provides more intuition for our general approach.

3. **Other minor changes:**
     * We have incorporated all minor revisions to notation suggested by reviewers (in sections 2, 4, and 6).
     * The visualization of embedding weights in Figure 1 have been slightly revised.

---

### Meta-Review · Area_Chair_phVZ · 2023-12-14

**Metareview:**

The paper studies representation that is obtained by a simple feedforward neural network by maximizing the margin for arithmetic tasks such as modular addition, parity, and other finite group operations. It is shown that the Fourier features are obtained for modular addition and irreducible group representation of the input data is obtained for learning symmetric group, which supports the existing empirical results.

The result in this paper is quite interesting. It gives a deep insight on the problem. The theoretical result obtained in this paper matches the empirical results almost perfectly. Then, I recommend acceptance for this paper.

**Justification For Why Not Higher Score:**

While this paper shows a very interesting result, its generality is a bit restrictive.

**Justification For Why Not Lower Score:**

The result shown in this paper is technically strong and a kind of surprising. It is worth spotlight.

---

### Decision · Program_Chairs · 2024-01-16

Accept (spotlight)